# Adversarial Attacks on Linear Contextual Bandits

**Evrard Garcelon**[*]
Facebook AI Research
evrard@fb.com

**Baptiste Rozière**[*]
Facebook AI Research
broz@fb.com

**Laurent Meunier**[*]
Facebook AI Research
laurentmeunier@fb.com

**Jean Tarbouriech**
Facebook AI Research
jtarbouriech@fb.com

**Olivier Teytaud**
Facebook AI Research
oteytaud@fb.com

**Alessandro Lazaric**
Facebook AI Research
lazaric@fb.com

**Matteo Pirotta**
Facebook AI Research
pirotta@fb.com

## Abstract

Contextual bandit algorithms are applied in a wide range of domains, from advertising to recommender systems, from clinical trials to education. In many of these domains, malicious agents may have incentives to force a bandit algorithm into a desired behavior. For instance, an unscrupulous ad publisher may try to increase their own revenue at the expense of the advertisers; a seller may want to increase the exposure of their products, or thwart a competitor's advertising campaign. In this paper, we study several attack scenarios and show that a malicious agent can force a linear contextual bandit algorithm to pull any desired arm $T - o(T)$ times over a horizon of $T$ steps, while applying adversarial modifications to either rewards or contexts with a cumulative cost that only grow logarithmically as $O(\log T)$. We also investigate the case when a malicious agent is interested in affecting the behavior of the bandit algorithm in a single context (e.g., a specific user). We first provide sufficient conditions for the feasibility of the attack and an efficient algorithm to perform an attack. We empirically validate the proposed approaches in synthetic and real-world datasets.

## 1   Introduction

Recommender systems are at the heart of the business model of many industries like e-commerce or video streaming [1, 2]. The two most common approaches for this task are based either on matrix factorization [3] or bandit algorithms [4], which both rely on a unaltered feedback loop between the recommender system and the user. In recent years, a fair amount of work has been dedicated to understanding how targeted perturbations in the feedback loop can fool a recommender system into recommending low quality items.

Following the line of research on adversarial attacks in supervised learning [5, 6, 7, 8, 9], attacks on recommender systems have been focused on filtering-based algorithms [10, 11] and offline contextual bandits [12]. The question of adversarial attacks for online bandit algorithms has only been studied quite recently [13, 14, 15, 16], and solely in the multi-armed stochastic setting. Although the idea of online adversarial bandit algorithms is not new (see EXP3 algorithm in [17]), the focus is different from what we are considering in this article. Indeed, algorithms like EXP3 or EXP4 [18] are designed to find optimal actions in hindsight in order to adapt to any rewards stream.

---

[*] indicates equal contribution

The opposition between adversarial and stochastic bandit settings has sparked interests in studying a middle ground. In [19], the learning algorithm has no knowledge of the type of feedback it receives (either stochastic or adversarial). In [20, 21, 22, 23, 24], the rewards are assumed to be corrupted by adversarial rewards. The authors focus on building algorithms able to find the optimal actions even in the presence of some non-random perturbations. This setting is different from what is studied in this article because those perturbations are bounded and agnostic to arms pulled by the learning algorithm, i.e., the adversary corrupt the rewards before the algorithm chooses an arm.

In the broader Deep Reinforcement Learning (DRL) literature, the focus is placed on modifying the observations of different states to fool a DRL system at inference time [25, 26] or the rewards [27].

**Contribution.** In this work, we first follow the research direction opened by [13] where the attacker has the objective of fooling a learning algorithm into taking a specific action as much as possible. For example in a news recommendation problem, as described in [4], a bandit algorithm chooses between $K$ articles to recommend to a user, based on some information about them, called context. We assume that an attacker sits between the user and the website, they can choose the reward (i.e., click or not) for the recommended article observed by the recommending algorithm. Their goal is to fool the bandit algorithm into recommending some articles to most users. The contributions of our work can be summarized as follows:

- We extend the work of [13, 14] to the contextual linear bandit setting showing how to perturb rewards for both stochastic and adversarial algorithms, forcing **any** bandit algorithms to pull a specific set of arms, $o(T)$ times for logarithmic cost for the attacker.

- We analyze, for the first time, the setting in which the attacker can only modify the context $x$ associated with the current user (the reward is not altered). The goal of the attacker is to fool the bandit algorithm into pulling arms of a target set for most users (i.e., contexts) while minimizing the total norm of their attacks. We show that the widely known LINUCB algorithm [28, 18] is vulnerable to this new type of attack.

- We present a harder setting for the attacker, where the latter can only modify the context associated to a specific user. This situation may occur when a malicious agent has infected some computers with a Remote Access Trojan (RAT). The attacker can then modify the history of navigation of a specific user and, as a consequence, the information seen by the online recommender system. We show how the attacker can attack the two very common bandit algorithms LINUCB and Linear Thompson Sampling (LINTS) [29, 30] and, in certain cases, force them to pull a set of arms most of the time when a specific context (i.e., user) is presented to the algorithm (i.e., visits a website).

## 2 Preliminaries

We consider the standard contextual linear bandit setting with $K \in \mathbb{N}$ arms. At each time $t$, the agent observes a context $x_t \in \mathbb{R}^d$, selects an action $a_t \in [\![1, K]\!]$ and observes a reward: $r_{t,a_t} = \langle \theta_{a_t}, x_t \rangle + \eta_{a_t}^t$ where for each arm $a$, $\theta_a \in \mathbb{R}^d$ is a feature vector and $\eta_{a_t}^t$ is a conditionally independent zero-mean, $\sigma^2$-subgaussian noise. The contexts are assumed to be sampled *stochastically* except in App. D.

**Assumption 1.** *There exist $L > 0$ and $\mathcal{D} \subset \mathbb{R}^d$, such that for all $t$, $x_t \in \mathcal{D}$ and, $\forall x \in \mathcal{D}, \forall a \in [\![1, K]\!]$, $\|x\|_2 \leq L$ and $\langle \theta_a, x \rangle \in (0, 1]$. In addition, we assume that there exists $S > 0$ such that $\|\theta_a\|_2 \leq S$ for all arms $a$.*

The agent minimizes the cumulative regret after $T$ steps $R_T = \sum_{t=1}^{T} \langle \theta_{a_t^\star}, x_t \rangle - \langle \theta_{a_t}, x_t \rangle$, where $a_t^\star := \mathrm{argmax}_a \langle \theta_a, x_t \rangle$. A bandit learning algorithm $\mathfrak{A}$ is said to be *no-regret* when it satisfies $R_T = o(T)$, i.e., the average expected reward received by $\mathfrak{A}$ converges to the optimal one. Classical bandit algorithms (e.g., LINUCB and LINTS) compute an estimate of the unknown parameters $\theta_a$ using past observations. Formally, for each arm $a \in [K]$ we define $S_a^t$ as the set of times up to $t - 1$ (included) where the agent played arm $a$. Then, the estimated parameters are obtained through regularized least-squares regression as $\widehat{\theta}_a^t = (X_{t,a} X_{t,a}^\top + \lambda I)^{-1} X_{t,a} Y_{t,a}$, where $\lambda > 0$, $X_{t,a} = (x_i)_{i \in S_a^t} \in \mathbb{R}^{d \times |S_a^t|}$ and $Y_{t,a} = (r_{i,a_i})_{i \in S_a^t} \in \mathbb{R}^{|S_a^t|}$. Denote by $V_{t,a} = \lambda I + X_{t,a} X_{t,a}^\top$ the design matrix of the regularized least-square problem and by $\|x\|_V = \sqrt{x^\top V x}$ the weighted norm

w.r.t. any positive matrix $V \in \mathbb{R}^{d \times d}$. We define the confidence set:

$$\mathcal{C}_{t,a} = \left\{ \theta \in \mathbb{R}^d \; : \; \left\| \theta - \widehat{\theta}_{t,a} \right\|_{V_{t,a}} \leq \beta_{t,a} \right\} \tag{1}$$

where $\beta_{t,a} = \sigma \sqrt{d \log \left( (1 + L^2(1 + |S_a^t|)/\lambda)/\delta \right)} + S\sqrt{\lambda}$, which guarantees that $\theta_a \in \mathcal{C}_{t,a}$, for all $t > 0$, w.p. $1 - \delta$. This uncertainty is used to balance the exploration-exploitation trade-off either through optimism (e.g., LINUCB) or through randomization (e.g., LINTS).

## 3  Online Adversarial Attacks on Rewards

The ultimate goal of a malicious agent is to force a bandit algorithm to perform a desired behavior. An attacker may simply want to induce the bandit algorithm to perform poorly—ruining the users' experience—or to force the algorithm to suggest a specific arm. The latter case is particularly interesting in advertising where a seller may want to increase the exposure of its product at the expense of the competitors. Note that the users' experience is also compromised by the latter attack since the suggestions they will receive will not be tailored to their needs. Similarly to [14, 13], we focus on the latter objective, i.e., to fool the bandit algorithm into pulling arms in $A^\dagger$, a set of target arms, for $T - o(T)$ time steps (*independently of the user*).

A way to obtain this behavior is to dynamically modify the reward in order to make the bandit algorithm believe that $a^\dagger$ is optimal, for some $a^\dagger \in A^\dagger$. Clearly, the attacker has to pay a price in order to modify the perceived bandit problem and fool the algorithm. If there is no restriction on when and how the attacker can alter the reward, the attacker can easily fool the algorithm. However, this setting is not interesting since the attacker may pay a cost higher than the loss suffered by the attacked algorithm. An attack strategy is considered successful when the total cost of the attack is sublinear in $T$.

In this section, we show that under Assumption 1, there exists an attack algorithm that is successful against any bandit algorithm, stochastic or adverserial.

**Setting.** We assume that the attacker has the same knowledge as the bandit algorithm $\mathfrak{A}$ about the problem (i.e., knows $\sigma$ and $L$). The attacker is assumed to be able to observe the context $x_t$, the arm $a_t$ pulled by $\mathfrak{A}$, and can modify the reward received by $\mathfrak{A}$. When the attacker modifies the reward $r_{t,a_t}$ into $\widetilde{r}_{t,a_t}$ the *instantaneous* cost of the attack is defined as $c_t := \left| r_{t,a_t} - \widetilde{r}_{t,a_t} \right|$. The goal of the attacker is to fool algorithm $\mathfrak{A}$ such that the arms in $A^\dagger$ are pulled $T - o(T)$ times and $\sum_{t=1}^T c_t = o(T)$. We also assume that the action for the arms in the target set is strictly positive for every context $x \in \mathcal{D}$. That is to say that $\Delta := \min_{x \in \mathcal{D}} \left\{ \langle x, \theta_{a_\star^\dagger(x)} \rangle - \max_{a \in A^\dagger, a \neq a_\star^\dagger(x)} \langle x, \theta_a \rangle \right\} > 0$ where $a_\star^\dagger(x) = \arg\max_{a \in A^\dagger} \langle x, \theta_a \rangle$ for every $x \in \mathcal{D}$.

**Attack idea.** We leverage the idea presented in [14] and [13] where the attacker lowers the reward of arms $a \notin A^\dagger$ so that algorithm $\mathfrak{A}$ learns that an arm of the target set is optimal for every context. Since $\mathfrak{A}$ is assumed to be no-regret, the attacker only needs to modify the rewards $o(T)$ times to achieve this goal. Lowering the rewards has the effect of shifting the vectors $(\theta_a)_{a \notin A^\dagger}$ to new vectors $(\theta_a')_{a \notin A^\dagger}$ such that for all arms $a \notin A^\dagger$ and all contexts $x \in \mathcal{D}$, there exists an arm $a^\dagger \in A^\dagger$ such that $\langle \theta_a', x \rangle \leq \langle \theta_{a^\dagger}, x \rangle$. Since rewards are assumed to be bounded (see Asm. 1), this objective can be achieved by simply forcing the reward of non-target arms $a \notin A^\dagger$ to the minimum value. Contextual ACE (see Fig. 2) implements a soft version of this idea by leveraging the knowledge of the reward distribution. At each round $t$, Contextual ACE modifies the reward perceived by $\mathfrak{A}$ as follows:

$$\widetilde{r}_{t,a_t}^1 = \eta_t' \mathbb{1}_{\{a_t \notin A^\dagger\}} + r_{t,a_t} \mathbb{1}_{\{a_t \in A^\dagger\}} \tag{2}$$

where $\eta_t'$ is a $\sigma$-subgaussian random variable generated by the attacker independently of all other random variables. Contextual ACE transforms the original problem into a *stationary* bandit problem in which there is a targeted arm that is optimal for all contexts and all non targeted arms have expected reward of 0. The following propostion shows that the cumulative cost of the attack is sublinear.

**Proposition 1.** *For any $\delta \in (0, 1/K]$, when using Contextual ACE algorithm (Fig. 1) with perturbed rewards $\widetilde{r}^1$, with probability at least $1 - K\delta$, algorithm $\mathfrak{A}$ pulls an arm in $A^\dagger$ for $T - o(T)$ time steps and the total cost of attacks is $o(T)$.*

The proof of this proposition is provided in App. A.1. While Prop. 1 holds for any no-regret algorithm $\mathfrak{A}$, we can provide a more precise bound on the total cost by inspecting the algorithm. For example, we can show (see App. E), that, with probability at least $1 - K\delta$, the number of times LINUCB [28] pulls arms not in $A^\dagger$ is at most $\sum_{j \notin A^\dagger} N_j(T) \leq \frac{64K\sigma^2\lambda S^2}{\Delta^2} \left( d\log\left(\frac{\lambda + \frac{TL^2}{d}}{\delta^2}\right) \right)^2$. This directly translates into a bound on the total cost.

**Comparison with ACE [14].** In the stochastic setting, the ACE algorithm [14] leverages a bound on the expected reward of each arm in order to modify the reward. However, the perturbed reward process seen by algorithm $\mathfrak{A}$ is non-stationary and in general there is no guarantee that an algorithm minimizing the regret in a stationary bandit problem keeps the same performance when the bandit problem is not stationary anymore. Nonetheless, transposing the idea of the ACE algorithm to our setting would give an attack of the following form, where at time $t$, Alg. $\mathfrak{A}$ pulls arm $a_t$ and receives rewards $\widetilde{r}^2_{t,a_t}$:

$$\widetilde{r}^2_{t,a_t} = (r_{t,a_t} + \max(-1, \min(0, C_{t,a_t})))\mathbb{1}_{\{a_t \notin A^\dagger\}} + r_{t,a_t}\mathbb{1}_{\{a_t \in A^\dagger\}}$$

with $C_{t,a_t} = (1 - \gamma)\min_{a^\dagger \in A^\dagger} \min_{\theta \in \mathcal{C}_{t,a^\dagger}} \langle \theta, x_t \rangle - \max_{\theta \in \mathcal{C}_{t,a_t}} \langle \theta, x_t \rangle$. Note that $\mathcal{C}_{t,a}$ is defined as in Eq. 1 using the *non-perturbed* rewards, i.e., $Y_{t,a} = (r_{i,a_i})_{i \in S^t_a}$.

**Bounded Rewards.** The bounded reward assumption is necessary in our analysis to prove a formal bound on the total cost of the attacks for *any* no-regret bandit algorithm, otherwise we need more information about the attacked algorithm. In practice, the second attack on the rewards, $\widetilde{r}^2$, can be used in the case of unbounded rewards for any algorithms. The difficulty for unbounded reward is that the attacker has to adapt to the environment reward but in order to do so the reward process observed by the bandit algorithm becomes non-stationary under the attack. Thus, there is no guarantee that an algorithm like LINUCB will pull a target arm as the proof relies on the environment observed by the bandit algorithm being stationary. We observe empirically that the total cost of attack is sublinear when using $\widetilde{r}^2$.

[13] does not assume that rewards are bounded but focus on attacking algorithms in the stochastic multi-armed setting. That is to say they study attacks only designed for $\varepsilon$-greedy and UCB while we provide an efficient attack for any algorithms in the linear contextual case. We can extend their work, and thus remove the bounded reward assumption, in the linear contextual case by using the following attack, designed only for LINUCB:

$$\widetilde{r}^3_{t,a_t} = \left( r_{t,a_t} + \min_{a^\dagger \in A^\dagger} \min_{\theta \in \mathcal{C}_{t,a^\dagger}} \langle \theta, x_t \rangle - \max_{\theta \in \mathcal{C}_{t,a_t}} \langle \theta, x_t \rangle \right) \mathbb{1}_{\{a_t \notin A^\dagger\}} + r_{t,a_t}\mathbb{1}_{\{a_t \in A^\dagger\}} \tag{3}$$

with $C_{t,a}$ defined as in Eq. (1). Although, the attack $\widetilde{r}^3$ is not stationary, it is possible to prove that the total cost of attack is $\mathcal{O}(\log(T))$ because we know that the attacked bandit algorithm is LINUCB.

**Constrained Attack.** When the attacker has a constraint on the instantaneous cost of the attack, using the perturbed reward $\widetilde{r}^1$ may not be possible as the cost of the attack at time $t$ is not decreasing over time. Using the perturbed reward $\widetilde{r}^2$ offers a more flexible type of attack with more control on the instantaneous cost thanks to the parameter $\gamma$. But it still suffers from a minimal cost of attack from lowering rewards of arms not in $A^\dagger$.

**Defense mechanism.** The attack based on reward $\widetilde{r}_1$ is hardly detectable without prior knownledge about the problem. In fact, the reward process associated to $\widetilde{r}_1$ is stationary and compatible with the assumption about the true reward (e.g., subgaussian). While having very low rewards is reasonable in advertising, it can make the attack easily detectable in some other problems. On the other hand, the fact that $\widetilde{r}_2$ is a non-stationary process makes this attack easier to detect. When some data are already available on each arm, the learner can monitor the difference between the average rewards per action computed on new and old data.

## 4 Online Adversarial Attacks on Contexts

In this section, we consider the attacker to be able to alter the context $x_t$ perceived by the algorithm rather than the reward. The attacker is now restricted to change the type of users presented to the learning algorithm $\mathfrak{A}$, hence changing its perception of the environment. We show that under the assumption that the attacker knows a lower-bound to the reward of the target set, it is possible to fool LINUCB.

| **For** time $t = 1, 2, ..., T$ **do** |
| --- |
| 1. Alg. $\mathfrak{A}$ chooses arm $a_t$ based on context $x_t$ |
| 2. Environment generates reward: $r_{t,a_t} = \langle \theta_{a_t}, x_t \rangle + \eta_t$ with $\eta_{a_t}^t$ conditionally $\sigma^2$-subgaussian |
| 3. Attacker observes reward $r_{t,a_t}$ and feeds the perturbed reward $\widetilde{r}_{t,a_t}^1$ (or $\widetilde{r}_{t,a_t}^2$) to $\mathfrak{A}$ |

Figure 1: Contextual ACE algorithm

| **Input:** attack parameter: $\alpha$ |
| --- |
| **For** time $t = 1, 2, ..., T$ **do** |
| 1. Attacker observes the context $x_t$, computes potential arm $a_t'$ and sets $\widetilde{x}_t = x_t + (\alpha(x_t) - 1)x_t \, \mathbb{1}_{\{a_t' \notin A^\dagger\}}$ |
| 2. Alg. $\mathfrak{A}$ chooses arm $a_t$ based on context $\widetilde{x}_t$ |
| 3. Environment generates reward: $r_{t,a_t} = \langle \theta_{a_t}, x_t \rangle + \eta_t$ with $\eta_t$ conditionally $\sigma^2$-subgaussian |
| 4. Alg. $\mathfrak{A}$ observes reward $r_{t,a_t}$ |

Figure 2: ConicAttack algorithm.

**Setting.** As in Sec. 3, we consider the attacker to have the same knowledge about the problem as $\mathfrak{A}$. The main difference with the previous setting is that the attacker attacks before the algorithm. We adopt a *white-box* [6] setting attacking LINUCB. The goal of the attacker is unchanged: they aim at forcing the algorithm to pull arms in $A^\dagger$ for $T - o(T)$ time steps while paying a sublinear total cost. We denote by $\widetilde{x}_t$ the context after the attack and by $c_t = \|x_t - \widetilde{x}_t\|_2$ the instantaneous cost.

**Difference between attacks on contexts and rewards.** Perturbing contexts is fundamentally different from perturbing the rewards. The attacker only modifies the context that is shown to the bandit algorithm. The true context, which is used to compute the reward, remains unchanged. In other words, the attacker cannot modify the reward observed by the bandit algorithm. Instead, the attack algorithm described in this section fools the bandit algorithm by making the rewards appear small relative to the contexts and requires more assumptions on the bandit algorithm than in Sec. 3.

**Attack Idea.** The idea of the attack in this setting is similar to the attack of Sec. 3. The attacker builds a bandit problem where arm an $a^\dagger \in A^\dagger$ is optimal for all contexts by lowering the perceived value of all other arms not in $A^\dagger$. The attacker cannot modify the reward but, thanks to the linear reward assumption, they can scale the contexts to decrease the predicted rewards in the original context.

At time $t$, the attacker receives the context $x_t$ and computes the attack. Thanks to the white-box setting, it computes the arm $a_t$ that algorithm $\mathfrak{A}$ would pull if presented with context $x_t$. If $a_t \notin A^\dagger$ then the attacker changes the context to $\widetilde{x}_t = \alpha_{a_t} x_t$ with $\alpha_{a_t} > \max_{x \in \mathcal{D}} \min_{a^\dagger \in A^\dagger} \langle \theta_{a_t}, x \rangle / \langle \theta_{a^\dagger}, x \rangle$. This factor is chosen such that for a ridge regression computed on the dataset $(\alpha x_i, \langle \theta, x_i \rangle)_i$ outputs a parameter close to $\theta/\alpha$ therefore the attacker needs to choose $\alpha$ such that for every context $x \in \mathcal{D}$, $\langle x, \theta/\alpha \rangle \leq \max_{a^\dagger \in A^\dagger} \langle x, \theta_{a^\dagger} \rangle$. In other words, the attacker performs a dilation of the incoming context every time algorithm $\mathfrak{A}$ does not pull an arm in $A^\dagger$. The fact that the decision rule used by LINUCB is invariant by dilation guarantees that the attacker will not inadvertently lower the perceived rewards for arms in $A^\dagger$. Because the rewards are assumed to be linear, presenting a large context $\alpha x$ and receiving the reward associated with the normal context $x$ will skew the estimated rewards of LINUCB. The attack protocol is summarized in Fig. 2.

In order to compute the parameter $\alpha$ used in the attack, we make the following assumption concerning the performance of the arms in the target set:

**Assumption 2.** *For all $x \in \mathcal{D}$, there exists $a^\dagger \in A^\dagger$, such that $0 < \nu \leq \langle x, \theta_{a^\dagger} \rangle$ and $\nu$ is known to the attacker.*

**Knowing $\nu$.** For advertising and recommendation systems, knowing $\nu$ is not problematic. Indeed in those cases, the reward is the probability of impression of the ad ($r \in [0, 1]$). The attacker has the freedom to choose one of multiple target arms with strictly positive click probability in every context. This freedom is an important aspect for the attacker since it allows the attacker to cherry pick the target ad(s). In particular, the attacker can estimate $\nu$ based on data from previous campaigns (only for the target ad). For instance, a company could have run many ad campaigns for one of their products and try to get the defender's system to advertise it.

An issue is that the norm of the attacked context can be greater that the upper bound $L$ of Assumption 1. To prevent this issue, we choose a context-dependent multiplicative constant $\alpha(x) = \min\{2/\nu, L/\|x\|_2\}$ which amounts to clip the norm of the attacked context to $L$. In Sec. 6, we show that this attack is effective for different size of target arms sets. We also show that in the case of contexts such that $\|x\|_2 \leq \nu L/2$ that the cost of attacks is logarithmic in the horizon $T$.

**Proposition 2.** *Using the attack described in Fig. 2 and assuming that $\|x\|_2 \leq \nu L/2$ for all contexts $x \in \mathcal{D}$, for any $\delta \in (0, 1/K]$, with probability at least $1 - K\delta$, the number of times LINUCB does not*

*pull an arm in $A^\dagger$ before time $T$ is at most $\sum_{j \notin A^\dagger} N_j(T) \le 32K^2 \left( \frac{\lambda}{\alpha^2} + \sigma^2 d \log \left( \frac{\lambda d + TL^2\alpha^2}{d\lambda\delta} \right) \right)^3$ with $N_j(T)$ the number of times arm $j$ has been pulled during the first $T$ steps, The total cost for the attacker is bounded by: $\sum_{t=1}^{T} c_t \le \frac{64K^2}{\nu} \left( \frac{\lambda}{\alpha^2} + \sigma^2 d \log \left( \frac{\lambda d + TL^2\alpha^2}{d\lambda\delta} \right) \right)^3$ with $\alpha = 2/\nu$.*

The proof of Proposition 2 (see App. A.2) assumes that the attacker can attack at any time step, and that they can know in advance which arm will be pulled by Alg. $\mathfrak{A}$ in a given context. Thus it is not applicable to random exploration algorithms like LINTS [29] and $\varepsilon$-GREEDY. We also observed empirically that thowe two randomized algorithms are more robust to attacks (see Sec. 6) than LINUCB.

**Norm Clipping.** Clipping the norm of the attacked contexts is not beneficial for the attacker. Indeed, this means that an attacked context was violating the assumption (used by the bandit algorithm) that contexts are bounded by $L$. The attack could then be easily detectable and may succeed only because it is breaking an underlying assumption used by the bandit algorithm. Prop. 2 provides a theoretical grounding for the proposed attack when contexts are bounded by $\nu L/2$ and not only $L$. Although, we can not prove a bound on the cumulative cost of attacks in general, we show in Sec. 6 that attacks are still successful for multiple datasets where contexts are not bounded by $\nu L/2$.

## 5 Offline attacks on a Single Context

Previous sections focused on the man-in-the-middle (MITM) attack either on reward or context. The MITM attack allows the attacker to arbitrarily change the information observed by the recommender system at each round. This attack may be hardly feasible in practice, since the exchange channels are generally protected by authentication and cryptographic systems. In this section, we consider the scenario where the attacker has control over a single user $u$. As an example, consider the case where the device of the user is infected by a malware (e.g., Trojan horse), giving full control of the system to the malicious agent. The attacker can thus modify the context of the specific user (e.g., by altering the cookies) that is perceived by the recommender system. We believe that changes to the context (e.g., cookies) are more subtle and less easily detectable than changes to the reward (e.g., click). Moreover, if the reward is a purchase, it cannot be altered easily by taking control of the user's device. Clearly, the impact of the attacker on the overall performance of the recommender system depends on the frequency of the specific user, that is out of the attacker's control. It may be thus difficult to obtain guarantees on the cumulative regret of algorithm $\mathfrak{A}$. For this reason, we mainly focus on the study of the feasibility of the attack.

The attacker targets a specific user (i.e., the infected user) associated to a context $x^\dagger$. Similarly to Sec. 4, the objective of the attacker is to find the minimal change to the context presented to the recommender system $\mathfrak{A}$ such that $\mathfrak{A}$ selects an arm in $A^\dagger$. $\mathfrak{A}$ observes a modified context $\widetilde{x}$ instead of $x^\dagger$. After selecting an arm $a_t$, $\mathfrak{A}$ observes the true noisy reward $r_{t,a_t} = \langle \theta_{a_t}, x^\dagger \rangle + \eta_{a_t}^t$. We still study a white-box setting: the attacker can access all the parameters of $\mathfrak{A}$.

In this section, we show under which condition it is possible for an attacker to fool both an optimistic and posterior sampling algorithm.

### 5.1 Optimistic Algorithm: LINUCB

We consider the LINUCB algorithm which chooses the arm to pull by maximizing an upper-confidence bound on the expected reward. For each arm $a$ and context $x$, the UCB value is given by $\max_{\theta \in \mathcal{C}_{t,a}} \langle x, \theta \rangle = \langle x, \hat{\theta}_a^t \rangle + \beta_{t,a} \|x\|_{\widetilde{V}_{t,a}^{-1}}$ (see Sec. 2). The objective of the attacker is to force LINUCB to pull an arm in $A^\dagger$ once presented with context $x^\dagger$. This means to find a perturbation of context $x^\dagger$ that makes any arm in $A^\dagger$ the most optimistic arm. Clearly, we would like to keep the perturbation as small as possible to reduce the cost for the attacker and the probability of being detected. Formally, the attacker needs to solve the following *non-convex* optimization problem:

$$\min_{y \in \mathbb{R}^d} \quad \|y\|_2 \qquad \text{s.t} \qquad \max_{a \notin A^\dagger} \max_{\theta \in \widetilde{\mathcal{C}}_{t,a}} \langle x^\dagger + y, \theta \rangle + \xi \le \max_{a^\dagger \in A^\dagger} \max_{\theta \in \widetilde{\mathcal{C}}_{t,a^\dagger}} \langle x^\dagger + y, \theta \rangle \tag{4}$$

where $\xi > 0$ is a parameter of the attacker and $\widetilde{\mathcal{C}}_{t,a} := \left\{ \theta \mid \|\theta - \hat{\theta}_a^t\|_{\widetilde{V}_{t,a}} \le \beta_{t,a} \right\}$ is the confidence set constructed by LINUCB. We use the notation $\widetilde{\mathcal{C}}, \widetilde{V}$ to stress the fact that LINUCB observes only

the modified context. In contrast to Sec. 3 and 4, the attacker may not be able to force the algorithm to pull any of the target arms in $A^\dagger$. In other words, Problem 4 may not be feasible. However, we are able to characterize the feasibility of (4).

**Theorem 1.** *Problem* (4) *is feasible at time $t$ iff.*

$$\exists \theta \in \cup_{a^\dagger \in A^\dagger} \widetilde{\mathcal{C}}_{t,a^\dagger}, \ \theta \notin Conv\left(\cup_{a \notin A^\dagger} \widetilde{\mathcal{C}}_{t,a}\right) \tag{5}$$

The condition given by Theorem 1 says that this attack can be done when there exists a vector $x$ for which an arm in $A^\dagger$ is assumed to be optimal according to LINUCB. The condition mainly stems from the fact that optimizing a linear product on a convex compact set will reach its maximum on the edge of this set. In our case this set is the convex hull of the confidence ellipsoids of LINUCB. Although it is possible to use an optimization algorithm for this class of non-convex problems—e.g., DC programming [31]—they are still slow compared to convex algorithms. Therefore, we present a simple convex relaxation of the previous problem for a single target arm $a^\dagger \in A^\dagger$ that still enjoys some empirical performance compared to Problem (4). The final attack can then be computed as the minimum of the attacks obtained for each $a^\dagger \in A^\dagger$. The relaxed problem is the following for each $a^\dagger \in A^\dagger$:

$$\min_{y \in \mathbb{R}^d} \quad \|y\|_2 \qquad \text{s.t} \qquad \max_{a \neq a^\dagger, a \notin A^\dagger} \max_{\theta \in \mathcal{C}_{t,a}} \langle x^\dagger + y, \theta - \hat{\theta}^t_{a^\dagger} \rangle \leq -\xi \tag{6}$$

Since the RHS of the constraint in Problem (4) can be written as $\max_{\theta \in \mathcal{C}_{t,a^\dagger}} \langle \theta, x^\dagger + y \rangle$ for any $y$, the relaxation here consists in using $\langle \theta, x^\dagger + y \rangle$ as a lower-bound to this maximum for any $\theta \in \mathcal{C}_{t,a^\dagger}$.

For the relaxed Problem (6), the same type of reasoning as for Problem (4) gives that Problem (6) is feasible if and only if $\hat{\theta}_{a^\dagger}(t) \notin \text{Conv}\left(\bigcup_{a \neq a^\dagger, a \notin A^\dagger} \mathcal{C}_{t,a}\right)$.

If Condition (5) is not met, no arm $a^\dagger \in A^\dagger$ can be pulled by LINUCB. Indeed, the proof of Theorem 1 shows that the upper-confidence of every arm in $A^\dagger$ is always dominated by another arm for any context. In other words, if any arm in $A^\dagger$ is optimal for some contexts then the condition is satisfied a linear number of times for LINUCB (for formal proof of this fact see App. A.4).

## 5.2 Random Exploration Algorithm: LINTS

The previous subsection focused on LINUCB, however we can obtain similar guarantees for algorithms with random exploration such as LINTS. In this case, it is not possible to guarantee that a specific arm will be pulled for a given context because of the randomness in the arm selection process. The objective is to guarantee that an arm from $A^\dagger$ is pulled with probability at least $1 - \delta$. Similarly to the previous subsection, the problem of the attacker can be written as:

$$\min_{y \in \mathbb{R}^d} \quad \|y\| \qquad \text{s.t} \qquad \mathbb{P}\left(\exists a^\dagger \in A^\dagger, \forall a \notin A^\dagger, \langle x^\dagger + y, \widetilde{\theta}_a - \widetilde{\theta}_{a^\dagger} \rangle \leq -\xi\right) \geq 1 - \delta \tag{7}$$

where the $\widetilde{\theta}_a$ for different arms $a$ are independently drawn from a normal distribution with mean $\hat{\theta}_a(t)$ and covariance matrix $\upsilon^2 \bar{V}_a^{-1}(t)$ with $\upsilon = \sigma \sqrt{9d \ln(T/\delta)}$. Solving this problem is not easy and in general not possible, even for a single arm. For a given $x$ and arm $a$, the random variable $\langle x, \widetilde{\theta}_a \rangle$ is normally distributed with mean $\mu_a(x) := \langle \hat{\theta}_a(t), x \rangle$ and variance $\sigma_a^2(x) := \upsilon^2 \|x\|^2_{\bar{V}_a^{-1}(t)}$. We can then write $\langle x, \widetilde{\theta}_a \rangle = \mu_a(x) + \sigma_a(x) Z_a$ with $(Z_a)_a \sim \mathcal{N}(0, I_K)$. For the sake of clarity, we drop the variable $x$ when writing $\mu_a(x)$ and $\sigma_a(x)$.

Let's imagine (just for this paragraph) that $A^\dagger = \{a^\dagger\}$, then the constraint in Problem (7) becomes $\left[1 - \mathbb{E}_{Z_{a^\dagger}}\left(\Pi_{a \notin A^\dagger} \Phi\left(\frac{\sigma_{a^\dagger} Z_{a^\dagger} + \mu_{a^\dagger} - \mu_a}{\sigma_a}\right)\right)\right] \leq \delta$ where $\Phi$ is the cumulative distribution function of a normally distributed Gaussian random variable. Unfortunately, computing exactly this expectation is an open problem.

In the more general case where $|A^\dagger| \geq 1$, rewriting the constraints of Problem (7) is not possible. Following the idea of [14], for every single target arm $a^\dagger \in A^\dagger$, a possible relaxation of the constraint in Problem (7) is, to ensure that there exists an arm $a^\dagger \in A^\dagger$ such that for every arm $a \notin A^\dagger$, $1 - \Phi\left((\mu_{a^\dagger} - \mu_a - \xi)/(\sqrt{\sigma_a^2 + \sigma_{a^\dagger}^2})\right) \leq \frac{\delta}{K - |A^\dagger|}$, where $|A^\dagger|$ is the cardinal of $A^\dagger$. Thus the

relaxed version of the attack on LINTS for a single arm $a^\dagger$ is:

$$\min_{y\in\mathbb{R}^d}\|y\| \quad \text{s.t.} \quad \forall a \notin A^\dagger, \langle x^\dagger + y, \hat{\theta}_{a^\dagger} - \hat{\theta}_a\rangle - \xi \geq \nu\Phi^{-1}\left(1 - \frac{\delta}{K-|A^\dagger|}\right)\|x^\dagger + y\|_{\bar{V}_a^{-1}+\bar{V}_{a^\dagger}^{-1}} \quad (8)$$

Problem (8) is similar to Problem (6) as the constraint is also a Second Order Cone Program but with different parameters (see App. C). As in section 5.1, we compute the final attack as the minimum of the attacks computed for each arm in $A^\dagger$.

## 6  Experiments

In this section, we conduct experiments on the attacks on contextual bandit problems with simulated data and two real-word datasets: MovieLens25M [32] and Jester [33]. The synthetic dataset and the data preprocessing step are presented in App. B.1.

### 6.1  Attacks on Rewards

We study the impact of the reward attack for $4$ contextual algorithms: LINUCB, LINTS, $\varepsilon$-GREEDY and EXP4. As parameters, we use $L = 1$ for the maximal norm of the contexts, $\delta = 0.01$, $\upsilon = \sigma\sqrt{d\ln(t/\delta))/2}$, $\varepsilon_t = 1/\sqrt{t}$ at each time step $t$ and $\lambda = 0.1$. We choose only a *unique target arm* $a^\dagger$. For EXP4, we use $N = 10$ experts with $N - 2$ experts returning a random arm at each time, one expert choosing arm $a^\dagger$ every time and one expert returning the optimal arm for every context. With this set of experts the regret of bandits with expert advice is the same as in the contextual case. To test the performance of each algorithm, we generate $40$ random contextual bandit problems and run each algorithm for $T = 10^6$ steps on each. We report the average cost and regret for each of the $40$ problems. Figure 3 (Top) shows the attacked algorithms using the attacked reward $\tilde{r}^1$ (reported as "stationary CACE") and the rewards $\tilde{r}^2$ (reported as CACE).

These experiments show that, even though the reward process is non-stationary, usual stochastic algorithms like LINUCB can still adapt to it and pull the optimal arm for this reward process (which is arm $a^\dagger$). The true regret of the attacked algorithms is linear as $a^\dagger$ is not optimal for all contexts. In the synthetic case, for the algorithms attacked with the rewards $\tilde{r}^2$, over 1M iterations and $\gamma = 0.22$, the target arm is drawn more than $99.4\%$ of the time on average for every algorithm and more than $97.8\%$ of the time for the stationary attack $\tilde{r}^1$ (see Table 2 in App. B.2). The dataset-based environments (see Figure 3 (Left)) exhibit the same behavior: the target arm is pulled more than $94.0\%$ of the time on average for all our attacks on Jester and MovieLens and more than $77.0\%$ of the time in the worst case (for LINTS attacked with the stationary rewards) (see Table 2).

### 6.2  Attacks on Contexts

We now illustrate the effectiveness of the attack in Alg. 2. We study the behavior of attacked LINUCB, LINTS, $\varepsilon$-GREEDY with different size of target arms set ($|A^\dagger|/K \in \{0.3, 0.6, 0.9\}$ with $K$ the total number of arms). We test the performance of LINUCB with the same parameters as in the previous experiments. Yet since the variance is much smaller in this case, we generate a random problem and run 20 simulations for each algorithm. The target arms are chosen randomly and we use the exact lower-bound on the reward of those arms to compute $\nu$.

Table 1: Percentage of iterations for which the algorithm pulled an arm in the target set $A^\dagger$ (with a target set size of $0.3K$ arms) (**Left**) Online attacks using ContextualConic ($CC$) algorithm. Percentages are averaged over 20 runs of 1M iterations. (**Right**) Offline attacks with exact (Full) and Relaxed optimization problem. Percentages are averaged over 40 runs of 1M iterations.

| | Synthetic | Jester | Movilens | | Synthetic | Jester | MovieLens |
|---|---|---|---|---|---|---|---|
| LINUCB | 28.91% | 26.59% | 31.13% | LINUCB | 0.07% | 0.01% | 0.39% |
| CC LinUCB | 98.55% | 98.36% | 99.61% | LINUCB Relaxed | 13.76% | 97.81% | 4.09% |
| $\varepsilon$-GREEDY | 25.7% | 25.85% | 31.78% | LINUCB Full | 88.30% | 99.98% | 99.99% |
| CC $\varepsilon$-GREEDY | 89.71% | 99.85% | 99.92% | $\varepsilon$-GREEDY | 0.01% | 0.00% | 0.03% |
| LINTS | 27.2% | 26.10% | 33.24% | $\varepsilon$-GREEDY Full | 99.98% | 99.95% | 99.97% |
| CC LINTS | 30.93% | 97.26% | 98.82% | LINTS | 0.02% | 0.01% | 0.05% |
| | | | | LINTS Relaxed | 18.21% | 80.48% | 5.56% |

Table 1 (Left) shows the percentage of times an arm in $A^\dagger$, for $|A^\dagger| = 0.3K$, has been selected by the attacked algorithm. We see that, as expected, CC LINUCB reaches a ratio of almost 1, meaning the

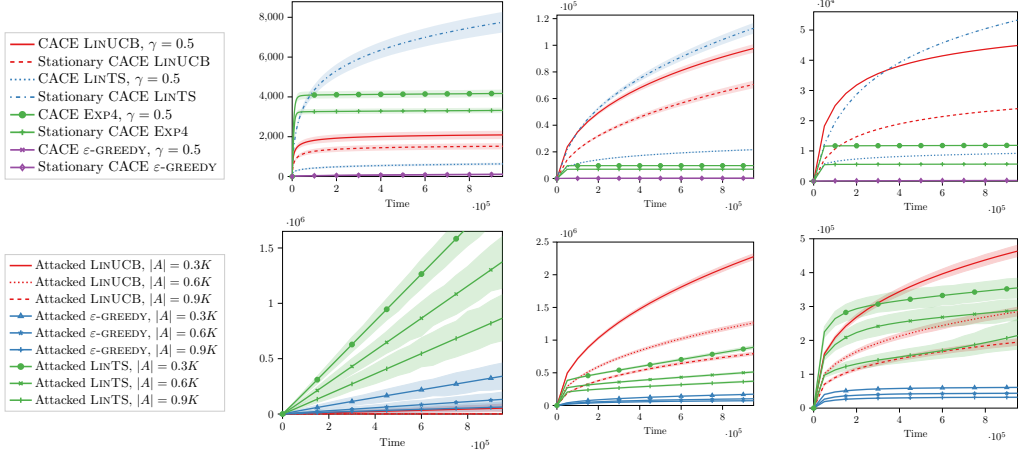

Figure 3: Total cost of attacks on rewards for the synthetic (Left, $\gamma = 0.22$), Jester (Center, $\gamma = 0.5$) and MovieLens (Right, $\gamma = 0.5$) environments. Bottom, total cost of ContextualConic attacks on the synthetic (Left), Jester (Center) and MovieLens (Right) environments.

target arms are indeed pulled a linear number of times. A more surprising result (at least not covered by the theory) is that $\varepsilon$-GREEDY exhibits the same behavior. Similarly to LINTS, $\varepsilon$-GREEDY exhibits some randomness in the action selection process. It can cause an arm $a^\dagger \in A^\dagger$ to be chosen when the context is attacked and interfere with the principle of the attack. We suspect that is what happens for LINTS. Fig. 3 (Bottom) shows the total cost of the attacks for the attacked algorithms . Despite the fact that the estimate of $\theta_{a^\dagger}$ can be polluted by attacked samples, it seems that LINTS can still pick up $a^\dagger$ as being optimal for some dataset like MovieLens and Jester but not on the simulated dataset.

## 6.3  Offline attacks on a Single Context

We now move to the setting described in Sec. 5 and test the same algorithms as in Sec. 6.2. We run 40 simulations for each algorithm and each attack type. The target context $x^\dagger$ is chosen randomly and the target arm as the arm minimizing the expected reward for $x^\dagger$. The attacker is only able to modify the incoming context for the target context (which corresponds to the context of one user) and the incoming contexts are sampled uniformly from the set of all possible contexts (of size 100). Table 1 (Right) shows the percentage of success for each attack. We observe that the non-relaxed attacks on $\varepsilon$-GREEDY and LINUCB work well across all datasets. However, the relaxed attack for LINUCB and LINTS are not as successful, on the synthetic dataset and MovieLens25M. The Jester dataset seems to be particularly suited to this type of attacks because the true feature vectors are well separated from the convex hull formed by the feature vectors of the other arms: only 5% of Jester's feature vectors are within the convex hull of the others versus 8% for MovieLens and 20% for the synthetic dataset. As expected, the cost of the attacks is linear on all the datasets (see Figure 6 in App. B.4). The cost is also lower for the non-relaxed than for the relaxed version of the attack on LINUCB. Unsurprisingly, the cost of the attacks on LINTS is the highest due to the need to guarantee that $a^\dagger$ will be chosen with high probability (95% in our experiments).

## 7   Conclusion

We presented several settings for online attacks on contextual bandits. We showed that an attacker can force any contextual bandit algorithm to almost always pull an arbitrary target arm $a^\dagger$ with only sublinear modifications of the rewards. When the attacker can only modify the contexts, we prove that LINUCB can still be attacked and made to almost always pull an arm in $A^\dagger$ by adding sublinear perturbations to the contexts. When the attacker can only attack a single context, we derive a feasibility condition for the attacks and we introduce a method to compute some attacks of small instantaneous cost for LINUCB, $\varepsilon$-GREEDY and LINTS. To the best of our knowledge, this paper is the first to describe effective attacks on the contexts of contextual bandit algorithms. Our numerical experiments, conducted on both synthetic and real-world data, validate our results and show that the attacks on all contexts are actually effective on several algorithms and with more permissible settings.

## Broader Impact

Adversarial attacks have been a major concerns in the machine learning community for some time [5, 6, 7, 8, 9] as they delve deeply into the robustness of such machine learning systems. Although, adversarial attacks have only been recently studied for bandits and reinforcement learning algorithms [12, 25]. Those settings are applied to a wide range of applications such as recommender systems or cooling down data centers [34].

In adversarial attacks on supervised algorithms and cryptography, it is well-accepted that the study and publication of attack schemes helps build trustful secure systems [35]. While there is a risk that our methods could be used by malicious attackers, we believe that they will also prompt some practitioners to ensure such modifications of the rewards or contexts of their data can be detected or even prevented.

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
