[Supplementary Material]

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

# A Proofs

In this appendix, we present the proofs of different theoretical results presented in the paper.

## A.1 Proof of Proposition 1

**Proposition.** *For any $\delta \in (0, 1/K]$, when using Contextual ACE algorithm (Alg. 1) with perturbed rewards $\tilde{r}^1$, with probability at least $1 - K\delta$, algorithm $\mathfrak{A}$ pulls an arm in $A^\dagger$ for $T - o(T)$ time steps and the total cost of attacks is $o(T)$.*

*Proof.* Let us consider the contextual bandit problem $\mathcal{A}_1$, with $K$ arms with contexts $x \in \mathcal{D}$ such that every arm in $a^\dagger \in A^\dagger$ has mean reward $\langle \theta_{a^\dagger}, x \rangle$ and all other arms has mean 0. Then the regret of algorithm $\mathfrak{A}$ for this bandit problem is upper-bounded with probability at least $1 - \delta$ by a function $f_{\mathfrak{A}}(T)$ such that $f_{\mathfrak{A}}(T) = o(T)$. In addition, the reward process fed to Alg. $\mathfrak{A}$ by the attacker is a stationary reward process with $\sigma^2$-subgaussian noise. Therefore, the number of times algorithm $\mathfrak{A}$ pulls an arm not in $A^\dagger$ is upper-bounded by $f_{\mathfrak{A}}(T)/\min_{x \in \mathcal{D}} \Delta(x)$ where for every context $x \in \mathcal{D}$, let $a_\star^\dagger(x) := \arg\max_{a \in A^\dagger} \langle x, \theta_a \rangle$ and $\Delta(x) = \langle x, \theta_{a_\star^\dagger(x)} \rangle - \max_{a \in A^\dagger, a \neq a_\star^\dagger(x)} \langle x, \theta_a \rangle$.

In addition, the total cost of the attack is upper-bounded by $\max_{a \in [\![1,K]\!]} \max_{x \in \mathcal{D}} |\langle x, \theta_a \rangle| (T - N_{A^\dagger}(T))$ where $N_{A^\dagger}(T)$ is the number of times an arm in $A^\dagger$ has been pulled up to time $T$. Thanks to the previous argument, $T - N_{A^\dagger}(T) \leq f_{\mathfrak{A}}(T)/\min_{x \in \mathcal{D}} \Delta(x)$. □

## A.2 Proof of Proposition 2

**Proposition.** *Using the attack described in Alg. 2, for any $\delta \in (0, 1/K]$, with probability at least $1 - K\delta$, the number of times LINUCB does not pull an arm in $A^\dagger$ is at most:*

$$\sum_{j \notin A^\dagger} N_j(T) \leq 32K^2 \left( \frac{\lambda}{\alpha^2} + \sigma^2 d \log \left( \frac{\lambda d + TL^2\alpha^2}{d\lambda\delta} \right) \right)^3$$

*with $N_j(T)$ the number of times arm $j$ has been pulled after $T$ steps, $||\theta_a|| \leq S$ for all arms $a$, $\lambda$ the regularization parameter of LINUCB and for all $x \in \mathcal{D}$, $||x||_2 \leq L$. The total cost for the attacker is bounded by:*

$$\sum_{t=1}^{T} c_t \leq \frac{64K^2}{\nu} \left( \frac{\lambda}{\alpha^2} + \sigma^2 d \log \left( \frac{\lambda d + TL^2\alpha^2}{d\lambda\delta} \right) \right)^3$$

*Proof.* Let $a_t$ be the arm pulled by LINUCB at time $t$. For each arms $a$, let $\tilde{\theta}_a(t)$ be the result of the linear regression with the attacked context and $\hat{\theta}_a(t, \lambda/\alpha^2)$ the one with the unattacked context and a regularization of $\frac{\lambda}{\alpha^2}$. At any time step $t$, we can write, for all $a \notin A^\dagger$:

$$
\begin{aligned}
\tilde{\theta}_a(t) &= \left( \lambda I_d + \sum_{l=0, a_l=a}^{t} \alpha^2 x_l x_l^\mathsf{T} \right)^{-1} \sum_{k=0, a_k=a}^{t} r_k \alpha x_k \\
&= \frac{1}{\alpha} \left( \frac{\lambda}{\alpha^2} I_d + \sum_{k=0, a_k=a}^{t} x_k x_k^\mathsf{T} \right)^{-1} \sum_{k=0, a_k=a}^{t} r_k x_k \\
&= \frac{\hat{\theta}_a(t, \lambda/\alpha^2)}{\alpha}
\end{aligned}
$$

We also note that, since the contexts are not modified for arms in $a^\dagger \in A^\dagger$: $\tilde{\theta}_{a^\dagger}(t) = \hat{\theta}_{a^\dagger}(t, \lambda)$. In addition, for any context $x$ and arm $a \notin A^\dagger$, the exploration term used by LINUCB becomes:

$$||x||_{\tilde{V}_{a,t}^{-1}} = \frac{1}{\alpha} ||x||_{\hat{V}_{a,t}^{-1}} \tag{9}$$

where $\tilde{V}_{a,t} = \lambda I_d + \sum_{l=0,a_l=a}^{t} \alpha^2 x_l x_l^\mathsf{T}$ and $\hat{V}_{a,t}^{-1} = \lambda/\alpha^2 I_d + \sum_{k=0,a_k=a}^{t} x_k x_k^\mathsf{T}$. For a time $t$, if presented with context $x_t$ LINUCB pulls arm $a_t \notin A^\dagger$, we have:

$$\alpha \left( \left\langle \hat{\theta}_{a^\dagger}(t), x_t \right\rangle + \beta_{a^\dagger}(t) ||x_t||_{V_{a^\dagger,t}^{-1}} \right) \leq \left\langle \hat{\theta}_{a_t}(t, \lambda/\alpha^2), x_t \right\rangle + \beta_{a_t}(t) ||x_t||_{\hat{V}_{a_t,t}^{-1}}$$

As $\alpha = \frac{2}{\nu} \geq \min_{a^\dagger \in A^\dagger} \frac{2}{\langle \theta_{a^\dagger}, x_t \rangle}$, we deduce that on the event that the confidence sets (Theorem 2 in [28]) hold for arm $a^\star$:

$$2 \leq \left\langle \hat{\theta}_{a_t}(t, \lambda/\alpha^2), x_t \right\rangle + \beta_{a_t}(t)||x_t||_{\hat{V}_{a_t,t}^{-1}} \leq \langle \theta_{a_t}, x_t \rangle + 2\beta_{a_t}(t)||x_t||_{\hat{V}_{a_t,t}^{-1}}$$

Thus, $1 \leq 2 - \langle \theta_{a_t}, x_t \rangle \leq 2\beta_{a_t}(t)||x_t||_{\hat{V}_{a_t,t}^{-1}}$. Therefore,

$$\sum_{t=1}^{T} \mathbb{1}_{\{a_t \notin A^\dagger\}} \leq \sum_{t=1}^{T} \min(2\beta_{a_t}(t)||x_t||_{\hat{V}_{a_t,t}^{-1}}, 1) \mathbb{1}_{\{a_t \notin A^\dagger\}}$$

$$\leq \sum_{j \notin A^\dagger} 2\beta_j(T) \sqrt{\sum_{t=1}^{T} \mathbb{1}_{\{a_t=j\}} \sum_{t=1,a_t=j}^{T} \min(1, ||x_t||_{\hat{V}_{j,t}^{-1}}^2)}$$

But using Lemma 11 from [28] and the bound on the $\beta_j(T)$ for all arms $j$, we have with Jensen inequality:

$$\sum_{t=1}^{T} \mathbb{1}_{\{a_t \notin A^\dagger\}} \leq 4 \sqrt{K \sum_{t=1}^{T} \mathbb{1}_{\{a_t \notin A^\dagger\}} d \log \left(1 + \frac{\alpha^2 T L^2}{\lambda d}\right)}$$

$$\times \left( \sqrt{\lambda/\alpha^2} S + \sigma \sqrt{2\log(1/\delta) + d\log(1 + \frac{\alpha^2 T L^2}{\lambda d})} \right)$$

$\square$

### A.3 Proof of Theorem 1

**Theorem.** *For any $\xi > 0$, Problem (4) is feasible if and only if:*

$$\exists \theta \in \bigcup_{a^\dagger \in A^\dagger} \mathcal{C}_{t,a^\dagger}, \qquad \theta \notin Conv \left( \bigcup_{a \notin A^\dagger} \mathcal{C}_{t,a} \right) \tag{10}$$

*where for every arm $a$, $\mathcal{C}_{t,a} := \{\theta \mid ||\theta - \hat{\theta}_a(t)||_{\tilde{V}_{a,t}} \leq \beta_a(t)\}$ with $\hat{\theta}_a(t)$ the least squares estimate for arm $a$ built by LINUCB and*

$$\tilde{V}_{a,t} = \lambda I_d + \sum_{l=1, x_l \neq x^\dagger}^{t} \mathbb{1}_{\{a_l=a\}} x_l x_l^\mathsf{T} + \sum_{l=1, x_l=x^\dagger}^{t} \mathbb{1}_{\{a_l=a\}} \tilde{x}_l \tilde{x}_l^\mathsf{T}$$

*the design matrix of LINUCB at time $t$ for all arms $a$ (where $\tilde{x}_l$ is the modified context)*

*Proof.* The proof of Theorem 1 is decomposed in two parts.

First, let us assume that Equation (10) is satisfied. Then, let us define $a^\dagger \in A^\dagger$ such that $\theta \in \mathcal{C}_{t,a^\dagger} \setminus Conv \left( \bigcup_{a \notin A^\dagger} \mathcal{C}_{t,a} \right)$, then by the theorem of separation of convex sets applied to $\mathcal{C}_{t,a^\dagger}$ and $\{\theta\}$. There exists a vector $v$ and $c_1 < c_2$ such that for all $y \in Conv \left( \bigcup_{a \neq a^\dagger} \mathcal{C}_{t,a} \right)$:

$$\langle y, v \rangle \leq c_1 < c_2 \leq \langle \theta, v \rangle.$$

Hence, for $\xi > 0$ we have that for $\tilde{v} = \frac{\xi}{c_2 - c_1} v$ that:

$$\langle y, \tilde{v} \rangle + \xi \leq \langle \theta, \tilde{v} \rangle$$

So the problem is feasible.

Secondly, let us assume that an attack is feasible. Then there exists a vector $y$ such that:

$$\max_{a^\dagger \in A^\dagger} \max_{\theta \in \mathcal{C}_{t,a^\dagger}} \langle y, \theta \rangle > c_1 := \max_{a \notin A^\dagger} \max_{\theta \in \mathcal{C}_{t,a}} \langle y, \theta \rangle \tag{11}$$

Let us reason by contradiction. We assume that $\bigcup_{a \in A^\dagger} \mathcal{C}_{t,a^\dagger} \subset \text{Conv}\left(\bigcup_{a \notin A^\dagger} \mathcal{C}_{t,a}\right)$ and consider

$$\theta^* \in \bigcup_{a \in A^\dagger} \mathcal{C}_{t,a^\dagger} \text{ such that } \langle y, \theta^* \rangle = \max_{a^\dagger \in A^\dagger} \max_{\theta \in \mathcal{C}_{t,a^\dagger}} \langle y, \theta \rangle$$

As we assumed $\bigcup_{a \in A^\dagger} \mathcal{C}_{t,a^\dagger} \subset \text{Conv}\left(\bigcup_{a \notin A^\dagger} \mathcal{C}_{t,a}\right)$, there exists $n \in \mathbb{N}^\star$, $\lambda_1, \cdots, \lambda_n \geq 0$ and $\theta_1, \cdots, \theta_n \in \bigcup_{a \notin A^\dagger} \mathcal{C}_{t,a}$ such that

$$\theta^* = \sum_{i=1}^{n} \lambda_i \theta_i \text{ and } \sum_{i=1}^{n} \lambda_i = 1$$

Thus

$$\langle y, \theta^* \rangle = \sum_{i} \lambda_i \langle y, \theta_i \rangle \leq c_1 \sum_{i=1}^{n} \lambda_i = c_1 \tag{12}$$

We assumed that the problem is feasible, so $c_1 < \langle y, \theta^* \rangle$ according to Eq. 11. It contradicts Eq. 12. $\square$

### A.4 Condition of Sec. 5

Figure 4: Illustrative example of condition (5). The target arm is arm $3$ or $5$ and the dashed black line is the convex hull of the other confidence sets. The ellipsoids are the confidence sets $\mathcal{C}_{t,a}$ for each arm $a$. If we consider only arms $\{1, 2, 4, 5\}$, and we use $5$ as the target arm, the condition (5) is satisfied as there is a $\theta$ outside the convex hull of the other confidence sets. On the other hand, if we consider arms $\{1, 2, 3, 4\}$ and we use $3$ as the target arm, the condition is not satisfied anymore.

Let us assume that there is an arm in $a^\dagger \in A^\dagger$ which is optimal for some contexts. More formally, there exists a subspace $V \subset \mathcal{D}$ such that:

$$\forall x \in V, \exists a_\star^\dagger(x) \in A^\dagger, \forall a \in [\![1, K]\!] \setminus \{a_\star^\dagger(x)\} \qquad \langle x, \theta_{a_\star^\dagger(x)} \rangle > \langle x, \theta_a \rangle .$$

We also assume that the distribution of the contexts is such that, for all $t$, $\mu := \mathbb{P}(x_t \in V) > 0$. Then, the regret is lower-bounded in expectation by:

$$\mathbb{E}(R_T) = \mathbb{E}\left(\sum_{t=1}^{T} \mathbb{1}_{\{x_t \in V\}}\left(\left\langle x_t, \theta_{a_\star^\dagger(x_t)} - \theta_{a_t}\right\rangle\right)\right) \geq \mu m(T) \min_{x \in V} \max_{a \neq a_\star^\dagger(x)} \langle \theta_{a_\star^\dagger(x)} - \theta_a, x \rangle$$

where $m(T)$ is the expected number of times $t \leq T$ such that condition (5) is not met. LINUCB guarantees that $\mathbb{E}(R_T) \leq \mathcal{O}(\sqrt{T})$ for every $T$. Hence, $m(T) \leq \mathcal{O}\left(\frac{\sqrt{T}}{\mu \min_{x \in V} \max_{a \neq a_\star^\dagger(x)} \langle \theta_{a_\star^\dagger(x)} - \theta_a, x \rangle}\right)$.

This means that, in an unattacked problem, condition (5) is met $T - \mathcal{O}(\sqrt{T})$ times. On the other hand, when the algorithm is attacked the regret of LINUCB is not sub-linear as the confidence bound for the target arm is not valid anymore. Hence we cannot provide the same type of guarantees for the attacked problem.

# B Experiments

## B.1 Datasets and preprocessing

We present here the datasets used in the article and how we preprocess them for numerical experiments conducted in Section 6.

We consider two types of experiments, one on synthetic data with a contextual MAB problems with $K = 10$ arms such that for every arm $a$, $\theta_a$ is drawn from a folded normal distribution in dimension $d = 30$. We also use a finite number of contexts (10), each of them is drawn from a folded normal distribution projected on the unit circle multiplied by a uniform radius variable (i.i.d. across all contexts). Finally, we scale the expected rewards in $(0, 1]$ and the noise is drawn from a centered Gaussian distribution $\mathcal{N}(0, 0.01)$.

The second type of experiments is conducted in the real-world datasets Jester [33] and MovieLens25M [32]. Jester consists of joke ratings on a continuous scale from $-10$ to $10$ for $100$ jokes from a total of $73421$ users. We use the features extracted via a low-rank matrix factorization ($d = 35$) to represent the actions (i.e., the jokes). We consider a complete subset of $40$ jokes and $19181$ users . Each user rates all the $40$ jokes. At each time, a user is randomly selected from the $19181$ users and mean rewards are normalized in $[0, 1]$. The reward noise is $\mathcal{N}(0, 0.01)$. The second dataset we use is MovieLens25M. It contains $25000095$ ratings created by $162541$ users on $62423$ movies. We perform a low-rank matrix factorization to compute users features and movies features. We keep only movies with at least $1000$ ratings, which leave us with $162539$ users and $3794$ movies. At each time step, we present a random user, and the reward is the scalar product between the user feature and the recommend movie feature. All rewards are scaled to lie in $[0, 1]$ and a Gaussian noise $\mathcal{N}(0, 0.01)$ is added to the rewards.

## B.2 Attacks on Rewards

In this appendix, we present empirical evolution of the total cost and the number of draws for a unique target arm as a function of the attack parameter $\gamma$ for the Contextual ACE attack with perturbed rewards $\tilde{r}^2$ on generated data.

(a) Total cost        (b) Number of draws

Figure 5: Total cost of attacks and number of draws of the target arm at $T = 10^6$ as a function of $\gamma$ on synthetic data

Fig. 5 (left) shows that the total cost of attacks seems to be quite invariant w.r.t. $\gamma$ except when $\gamma \to 0$ because the difference between the target arm and the other becomes negligible. This is also depicted by the total number of draws (Fig. 5, Right) as the number of draws plummets when $\gamma \to 0$.

## B.3 Attacks on all Contexts

Fig. B.3 shows the regret for all the attacks. This figure shows that even though the total cost of attacks is linear for algorithms like LINTS in the synthetic dataset, the regret is linear. More generally, we observe that the regret is linear for all attacked algorithms on all datasets.

Table 2: Number of draws of the target arm $a^\dagger$ at $T = 10^6$, for the synthetic data, $\gamma = 0.22$ for the Contextual ACE algorithm and for the Jester and MovieLens datasets $\gamma = 0.5$.

|  | Synthetic | Jester | Movilens |
|---|---|---|---|
| LINUCB | $86,731.6$ | $23,548.16$ | $25,017.31$ |
| CACE LINUCB | $996,238.6$ | $921,083.69$ | $944,721.28$ |
| Stationary CACE LINUCB | $995,578.88$ | $862,095.67$ | $931,531.6$ |
| $\varepsilon$-GREEDY | $111,380.44$ | $21,911.54$ | $3,165.81$ |
| CACE $\varepsilon$-GREEDY | $999,812.92$ | $999,755.72$ | $999,776.82$ |
| Stationary CACE $\varepsilon$-GREEDY | $999,806.32$ | $999,615.98$ | $999,316.76$ |
| LINTS | $91,664.8$ | $23,398.3$ | $30,189.84$ |
| CACE LINTS | $998,997.04$ | $976,708.9$ | $990,250.67$ |
| Stationary CACE LINTS | $977,850.96$ | $784,715.62$ | $845,512.98$ |
| EXP4 | $93,860.4$ | $29,147.01$ | $17,985.78$ |
| CACE EXP4 | $992,793.36$ | $989,214.36$ | $936,230.4$ |
| Stationary CACE EXP4 | $993,673.24$ | $988,463.56$ | $934,304.23$ |

## B.4 Attack on a single context

The attacks are computed by solving the optimization problems 4 and 6 (Sec. 5). We choose the libraries according to their efficiency for each problem we need to solve. For Problem (6) and Problem (8) we use CVXPY [36] and the ECOS solver. For Problem (4) we use the SLSQP method from the Scipy optimize library [37] to solve the full LINUCB problem (Equation 4) and QUADPROG to solve the quadratic problem to attack $\varepsilon$-GREEDY.

(a) Synthetic data      (b) Jester Dataset      (c) MovieLens Dataset

Figure 6: Total cost of the attacks for the attacks one one context on our synthetic dataset, Jester and MovieLens. As expected, the total cost is linear.

## C   Problem (8) as a Second Order Cone (SOC) Program

Problem (6) and Problem (8) are both SOC programs. We can see the similarities between both problems as follows. Let us define for every arm $a \notin A^\dagger$, the ellipsoid:

$$\mathcal{C}'_{t,a} := \left\{ y \in \mathbb{R}^d \mid ||y - \hat{\theta}_a(t)||_{A_a^{-1}(t)} \leq \upsilon \Phi^{-1} \left( 1 - \frac{\delta}{K - |A^\dagger|} \right) \right\}$$

with $A_a(t) = \tilde{V}_a^{-1}(t) + \tilde{V}_{a^\dagger}^{-1}(t)$ with $\tilde{V}_a(t)$ and $\tilde{V}_{a^\dagger}(t)$ the design matrix built by LINTS and $\hat{\theta}_a(t)$ the least squares estimate of $\theta_a$ at time $t$. Therefore for an arm $a$, the constraint in Problem (8) can be written for any $y \in \mathbb{R}^d$ and some arm $a^\dagger \in A^\dagger$ as:

$$\left\langle x^\star + y, \hat{\theta}_{a^\dagger}(t) \right\rangle - \xi \geq \max_{z \in \mathcal{C}'_{t,a}} \langle z, x^\star + y \rangle$$

Indeed for any $x \in \mathbb{R}^d$,

$$\begin{aligned}
\max_{y \in \mathcal{C}'_{t,a}} \langle y, x \rangle &= \left\langle x, \hat{\theta}_a(t) \right\rangle + \upsilon \Phi^{-1} \left( 1 - \frac{\delta}{K - |A^\dagger|} \right) \times \max_{||A_a^{-1/2}(t)u||_2 \leq 1} \langle u, x \rangle \\
&= \left\langle x, \hat{\theta}_a(t) \right\rangle + \upsilon \Phi^{-1} \left( 1 - \frac{\delta}{K - |A^\dagger|} \right) \max_{||z||_2 \leq 1} \left\langle z, A_a^{1/2}(t)x \right\rangle \\
&= \left\langle x, \hat{\theta}_a(t) \right\rangle + \upsilon \Phi^{-1} \left( 1 - \frac{\delta}{K - |A^\dagger|} \right) \| A_a^{1/2}(t)x \|_2
\end{aligned}$$

Thus, the constraint is feasible if and only if:

$$\hat{\theta}_{a^\dagger}(t) \notin \mathrm{Conv} \left( \bigcup_{a \notin A^\dagger} \mathcal{C}'_{t,a} \right)$$

## D  Attacks on Adversarial Bandits

In the previous sections, we studied algorithms with sublinear regret $R_T$, i.e., mainly bandit algorithms designed for stochastic stationary environments. Adversarial algorithms like EXP4 do not provably enjoy a sublinear **stochastic** regret $R_T$ (as defined in the introduction) [1]. In addition, because this type of algorithms are, by design, robust to non-stationary environments, one could expect them to induce a linear cost on the attacker. In this section, we show that this is not the case for most contextual adversarial algorithms. Contextual adversarial algorithms are studied through the reduction to the bandit with expert advice problem. This is a bandit problem with $K$ arms where at every step, $N$ experts suggest a probability distribution over the arms. The goal of the algorithm is to learn which expert gets the best expected reward in hindsight after $T$ steps. The regret in this type of problem is defined as $R_T^{\mathrm{exp}} = \mathbb{E} \left( \max_{m \in [\![1,N]\!]} \sum_{t=1}^T \sum_{j=1}^K E_{m,j}^{(t)} r_{t,j} - r_{t,a_t} \right)$ where $E_{m,j}^{(t)}$ is the probability of selecting arm $j$ for expert $m$. In the case of contextual adversarial bandits, the experts first observe the context $x_t$ before recommending an expert $m$. Assuming the current setting with linear rewards, we can show that if an algorithm $\mathfrak{A}$, like EXP4, enjoys a sublinear regret $R_T^{\mathrm{exp}}$, then, using the Contextual ACE attack with either $\tilde{r}^1$ or $\tilde{r}^2$, the attacker can fool the algorithm into pulling arm $a^\dagger$ a linear number of times under some mild assumptions. However, attacking contexts for this type of algorithm is difficult because, even though the rewards are linear, the experts are not assumed to use a specific model for selecting an action.

**Proposition 3.** *Suppose an adversarial algorithm $\mathfrak{A}$ satisfies a regret $R_T^{\mathrm{exp}}$ of order $o(T)$ for any bandit problem and that there exists an expert $m^\star$ such that $T - \sum_{t=1}^T \mathbb{E} \left( E_{m^\star, a_{t,\star}^\dagger}^{(t)} \right) = o(T)$ with $a_{t,\star}^\dagger$ the optimal arim in $A^\dagger$ at time $t$. Then attacking alg. $\mathfrak{A}$ with Contextual ACE leads to pulling arm $a^\dagger$, $T - o(T)$ of times in expectation with a total cost of $o(T)$ for the attacker.*

*Proof.* Similarly to the proof of Proposition 1, let's define the bandit with expert advice problem, $\mathcal{A}_i$, such that at each time $t$ the reward vector is $(\tilde{r}_{t,a}^i)_a$ (with $i \in \{1, 2\}$). The regret of this

algorithm is: $\tilde{R}_T^{i,\exp} = \mathbb{E}\left(\max_{m\in[\![1,N]\!]}\sum_{t=1}^T E_m^{(t)}\tilde{r}_t^i - \tilde{r}_{t,a_t}^i\right) \in o(T)$. The regret of the learner is:

$\mathbb{E}\left(\max_{m\in[\![1,N]\!]}\sum_{t=1}^T E_m^{(t)}r_t - r_{t,a_t}\right)$ where $a_t$ are the actions taken by algorithm $\mathcal{A}_i$ to minimize $\tilde{R}_T^{i,\exp}$. Then we have:

$$\tilde{R}_T^{i,\exp} \geq \mathbb{E}\left(\sum_{t=1}^T\sum_{j=1}^K (E_{m^\star,j}^{(t)} - \mathbb{1}_{\{j=a_{t,\star}^\dagger\}})\tilde{r}_{t,j}^i + \sum_{t=1}^T \tilde{r}_{t,a_{t,\star}^\dagger}^i - \tilde{r}_{t,a_t}^i\right)$$

Therefore,

$$\mathbb{E}\left(\sum_{t=1}^T \tilde{r}_{t,a_{t,\star}^\dagger}^i - \tilde{r}_{t,a_t}^i\right) \leq \tilde{R}_T^{i,\exp} + \mathbb{E}\left(\sum_{t=1}^T\sum_{j=1}^K (\mathbb{1}_{\{j=a_{t,\star}^\dagger\}} - E_{m^\star,j}^{(t)})\tilde{r}_{t,j}^i\right)$$

$$\leq \tilde{R}_T^{i,\exp} + \mathbb{E}\left(\sum_{t=1}^T (1 - E_{m^\star,a_{t,\star}^\dagger}^{(t)})\tilde{r}_{t,j}^i\right)$$

$$\leq \tilde{R}_T^{i,\exp} + \mathbb{E}\left(\sum_{t=1}^T (1 - E_{m^\star,a_{t,\star}^\dagger}^{(t)})\right)$$

For strategy $i = 1$, we have:

$$\mathbb{E}\left(\sum_{t=1}^T \tilde{r}_{t,a_{t,\star}^\dagger}^1 - \tilde{r}_{t,a_t}^1\right) = \sum_{t=1}^T \mathbb{E}\left(r_{t,a_{t,\star}^\dagger} - \mathbb{1}_{\{a_t\in A^\dagger\}}\right) \geq \left(T - \mathbb{E}\left(\sum_{t=1}^T \mathbb{1}_{\{a_t=a_{t,\star}^\dagger\}}\right)\right)\Delta$$

where $\Delta := \min_{x\in\mathcal{D}}\left\{\langle\theta_{a^\dagger(x)}, x\rangle - \max_{a\in A^\dagger, a\neq a^\dagger(x)}\langle\theta_{a'}, x\rangle\right\}$ with $a^\dagger(x) := \arg\max_{a\in A^\dagger}\langle\theta_a, x\rangle$.
Then, as $\tilde{R}_T^{1,\exp} \in o(T)$ and $\mathbb{E}\left(\sum_{t=1}^T (1 - E_{m^\star,a_{t,\star}^\dagger}^{(t)})\right) \in o(T)$, we deduce that $\mathbb{E}(\sum_t \mathbb{1}_{\{a_t=a_{t,\star}^\dagger\}}) = T - o(T)$.

For strategy $i = 2$, and $\delta > 0$, let us denote by $E_\delta$ the event that all confidence intervals hold with probability $1 - \delta$. But on the event $E_\delta$, for a time $t$ where $a_t \neq a_{t,\star}^\dagger$ and such that $-1 \leq C_{t,a_t} \leq 0$:

$$\tilde{r}_{t,a_t}^2 = r_{t,a_t} + C_{t,a_t} = (1-\gamma)\min_{a^\dagger\in A^\dagger}\min_{\theta\in\mathcal{C}_{t,a^\dagger}}\langle\theta, x_t\rangle + \eta_{a_t,t} + \langle\theta_a, x_t\rangle - \max_{\theta\in\mathcal{C}_{t,a_t}}\langle\theta, x_t\rangle$$

$$\leq (1-\gamma)\langle\theta_{a_{t,\star}^\dagger}, x_t\rangle + \eta_{a_t,t}$$

when $C_{t,a_t} > 0$ (still on the event $E_\delta$):

$$\tilde{r}_{t,a_t}^2 = r_{t,a_t} \leq (1-\gamma)\langle\theta_{a_{t,\star}^\dagger}, x_t\rangle + \eta_{a_t,t}$$

because $C_{t,a_t} > 0$ means that $(1-\gamma)\langle\theta_{a_{t,\star}^\dagger}, x_t\rangle \geq (1-\gamma)\min_{a^\dagger\in A^\dagger}\min_{\theta\in\mathcal{C}_{t,a^\dagger}}\langle\theta, x_t\rangle \geq \max_{\theta\in\mathcal{C}_{t,a_t}}\langle\theta, x_t\rangle \geq \langle\theta_a, x_t\rangle$. But finally, when $C_{t,a_t} \leq -1$, $\tilde{r}_{t,a_t}^2 = r_{t,a_t} - 1 \leq \eta_{a_t,t} \leq (1-\gamma)\langle\theta_{a_{t,\star}^\dagger}, x_t\rangle + \eta_{a_t,t}$. But on the complementary event $E_\delta^c$, $\tilde{r}_{t,a_t}^2 \leq r_{t,a_t}$. Thus, given that the expected reward is assumed to be bounded in $(0,1]$ (Assumption 1):

$$\mathbb{E}\left(\sum_{t=1}^T \tilde{r}_{t,a_{t,\star}^\dagger}^2 - \tilde{r}_{t,a_t}^2\right) = \mathbb{E}\left(\sum_{t=1}^T (r_{t,a^\dagger} - \tilde{r}_{t,a_t}^2)\mathbb{1}_{\{a_t\neq a_{t,\star}^\dagger\}}\right)$$

$$\geq \mathbb{E}\left(\sum_{t=1}^T \min\{\gamma\min_{x\in\mathcal{D}}\langle x, \theta_{a_{t,\star}^\dagger}\rangle, \Delta\}\mathbb{1}_{\{a_t\neq a_{t,\star}^\dagger\}}\mathbb{1}_{\{E_\delta\}}\right) - T\delta$$

Finally, putting everything together we have:

$$\mathbb{E}\left(\sum_{t=1}^T \gamma\min_{x\in\mathcal{D}}\langle x, \theta_{a_{t,\star}^\dagger}\rangle\mathbb{1}_{\{a_t\neq a_{t,\star}^\dagger\}}\right) \leq \tilde{R}_T^{2,\exp} + \mathbb{E}\left(\sum_{t=1}^T (1 - E_{m^\star,a_{t,\star}^\dagger}^{(t)})\right) + \delta T\left(\min\{\gamma\min_{a^\dagger\in A^\dagger}\min_{x\in\mathcal{D}}\langle x, \theta_{a^\dagger}\rangle, \Delta\} + 1\right)$$

Hence, because $\tilde{R}_T^{1,\exp} = o(T)$ and $\mathbb{E}\left(\sum_{t=1}^{T}(1 - E_{m^\star, a^\dagger}^{(t)})\right) = o(T)$ we have that for $\delta \leq 1/T$, the expected number of pulls of the optimal arm in $A^\dagger$ is of order $o(T)$. In addition, the cost for the attacker is bounded by:

$$\mathbb{E}\left(\sum_{t=1}^{T} c_t\right) = \mathbb{E}\left(\sum_{t=1}^{T} \mathbb{1}_{\{a_t \neq a_{t,\star}^\dagger\}}\big| \max(-1, \min(C_{t,a_t}, 0))\big|\right) \leq \mathbb{E}\left(\sum_{t=1}^{T} \mathbb{1}_{\{a_t \neq a_{t,\star}^\dagger\}}\right)$$

$\square$

The proof is similar to the one of Prop. 1. The condition on the expert in Prop. 3 means that there exists an expert which believes an arm $a^\dagger \in A^\dagger$ is optimal most of the time. The adversarial algorithm will then learn that this expert is optimal. Algorithm EXP4 has a regret $R_T^{\exp}$ bounded by $\sqrt{2TK\log(N)}$, thus the total number of pulls of arms not in $A^\dagger$ is bounded by $\sqrt{2TK\log(M)}/\gamma$. This result also implies that for adversarial algorithms like EXP3 [17], the same type of attacks could be used to fool $\mathfrak{A}$ into pulling arms in $A^\dagger$ because the MAB problem can be seen as a reduction of the contextual bandit problem with a unique context and one expert for each arm.

## E   Contextual Bandit Algorithms

In this appendix, we present the different bandit algorithms studied in this paper. All algorithms we consider except EXP4 uses disjoint models for building estimate of the arm feature vectors $(\theta_a)_{a \in [\![1,K]\!]}$. Each algorithm (except EXP4) builds least squares estimates of the arm features.

---

**Algorithm 1** Contextual LINUCB

---

**Input:** regularization $\lambda$, number of arms $K$, number of rounds $T$, bound on context norms: $L$, bound on norms $\theta_a$: $D$
Initialize for every arm $a$, $\bar{V}_a^{-1}(t) = \frac{1}{\lambda}I_d$, $\hat{\theta}_a(t) = 0$ and $b_a(t) = 0$
**for** $t = 1, ..., T$ **do**
    Observe context $x_t$
    Compute $\beta_a(t) = \sigma\sqrt{d\log\left(\frac{1 + N_a(t)L^2/\lambda}{\delta}\right)}$ with $N_a(t)$ the number of pulls of arm $a$
    Pull arm $a_t = \text{argmax}_a\langle\hat{\theta}_a(t), x_t\rangle + \beta_a(t)||x_t||_{\bar{V}_a^{-1}(t)}$
    Observe reward $r_t$ and update parameters $\hat{\theta}_a(t)$ and $\bar{V}_a^{-1}(t)$ such that:

$$\bar{V}_{a_t}(t+1) = \bar{V}_{a_t}(t) + x_t x_t^\mathsf{T}, \quad b_{a_t}(t+1) = b_{a_t}(t) + r_t x_t, \quad \theta_{a_t}(t+1) = \bar{V}_{a_t}^{-1}(t+1)b_{a_t}(t+1)$$

**end for**

---

---

**Algorithm 2** Linear Thompson Sampling with Gaussian prior

---

**Input:** regularization $\lambda$, number of arms $K$, number of rounds $T$, variance $\upsilon$
Initialize for every arm $a$, $\bar{V}_a^{-1}(t) = \lambda I_d$ and $\hat{\theta}_a(t) = 0$, $b_a(t) = 0$
**for** $t = 1, ..., T$ **do**
    Observe context $x_t$
    Draw $\tilde{\theta}_a \sim \mathcal{N}(\hat{\theta}_a(t), \upsilon^2\bar{V}_a^{-1}(t))$
    Pull arm $a_t = \text{argmax}_{a \in [\![1,K]\!]}\left\langle\tilde{\theta}_a, x_t\right\rangle$
    Observe reward $r_t$ and update parameters $\hat{\theta}_a(t)$ and $\bar{V}_a^{-1}(t)$

$$\bar{V}_{a_t}(t+1) = \bar{V}_{a_t}(t) + x_t x_t^\mathsf{T}, \quad b_{a_t}(t+1) = b_{a_t}(t) + r_t x_t, \quad \theta_{a_t}(t+1) = \bar{V}_{a_t}^{-1}(t+1)b_{a_t}(t+1)$$

**end for**

---

## F   Semi-Online Attacks

[14] studies what they call the offline setting for adversarial attacks on stochastic bandits. They consider a setting where a bandit algorithm is successively updated with mini-batches of fixed size $B$.

---

**Algorithm 3** $\varepsilon$-GREEDY

---

**Input:** regularization $\lambda$, number of arms $K$, number of rounds $T$, exploration parameter $(\varepsilon)_t$
Initialize, for all arms $a$, $\bar{V}_a^{-1}(t) = \lambda I_d$ and $\hat{\theta}_a(t) = 0$, $\varepsilon_t = 1$, $b_a(t) = 0$
**for** $t = 1, ..., T$ **do**
    Observe context $x_t$
    With probability $\varepsilon_t$, pull $a_t \sim \mathcal{U}(\llbracket 1, K \rrbracket)$, or pull $a_t = \text{argmax}\langle \theta_a, x_t \rangle$
    Observe reward $r_t$ and update parameters $\hat{\theta}_a(t)$ and $\bar{V}_a^{-1}(t)$

$$\bar{V}_{a_t}(t+1) = \bar{V}_{a_t}(t) + x_t x_t^\mathsf{T}, \quad b_{a_t}(t+1) = b_{a_t}(t) + r_t x_t,$$
$$\theta_{a_t}(t+1) = \bar{V}_{a_t}^{-1}(t+1) b_{a_t}(t+1)$$

**end for**

---

**Algorithm 4** EXP4

---

**Input:** number of arms $K$, experts: $(E_m)_{m \in \llbracket 1, N \rrbracket}$, parameter $\eta$
Set $Q_1 = (1/N)_{j \in \llbracket 1, N \rrbracket}$
**for** $t = 1, ..., T$ **do**
    Observe context $x_t$ and probability recommendation $(E_m^{(t)})_{m \in \llbracket 1, N \rrbracket}$
    Pull arm $a_t \sim P_t$ where $P_{t,j} = \sum_{k=1}^N Q_{t,k} E_{j,k}^{(t)}$
    Observe reward $r_t$ and define for all arms $i$ $\hat{r}_{t,i} = 1 - \mathbb{1}_{\{a_t=i\}}(1-r_t)/P_{t,i}$
    Define $\tilde{X}_{t,k} = \sum_a E_{k,a}^{(t)} \hat{r}_{t,a}$
    Update $Q_{t+1,j} = \exp(\eta Q_{t,i})/\sum_{j=1}^N \exp(\eta Q_{t,j})$ for all experts $i$
**end for**

---

The attacker can tamper with some of the incoming mini-batches. More precisely, they can modify the context, the reward and even the arm that was pulled for any entry of the attacked mini-batches. The main difference between this type of attacks and the online attacks we considered in the main paper is that we do not assume that we can attack from the start of the learning process: the bandit algorithm may have already converged by the time we attack.

We can still study the cumulative cost for the attacker to change the mini-batch in order to fool a bandit algorithm to pull a target arm $a^\dagger$ (here we take $A^\dagger = \{a^\dagger\}$). Contrarily to [14], we call this setting semi-online. We first study the impact of an attacker on LINUCB where we show that, by modifying only $(K-1)d$ entries from the batch $\mathcal{B}$, the attacker can force LINUCB to pull arm $a^\dagger$, $M'B - o(M'B)$ times with $M'$ the number of remaining batches updates. The cost of our attack is $\sqrt{MB}$ with $M$ the total number of batches.

**Cost of an attack:** If presented with a mini-batch $\mathcal{B}$, with elements $(x_t, a_t, r_t)$ composed of the context $x_t$ presented at time $t$, the action taken $a_t$ and the reward received $r_t$, the attacker modifies element $i$, namely $(x_t^i, a_t^i, r_t^i)$ into $(\tilde{x}_t^i, \tilde{a}_t^i, \tilde{r}_t^i)$. The cost of doing so is $c_t^i = ||x_t^i - \tilde{x}_t^i||_2 + |\tilde{r}_t^i - r_t^i| + \mathbb{1}_{\{a_t^i \neq \tilde{a}_t^i\}}$ and the total cost for mini-batch $\mathcal{B}$ is defined as $c_\mathcal{B} = \sum_{i \in \mathcal{B}} c_t^i$. Finally, we consider the cumulative cost of the attack over $M$ different mini-batches $\mathcal{B}_1, \ldots, \mathcal{B}_M$, $\sum_{l=1}^M c_{\mathcal{B}_l}$. The interaction between the environment, the attacker and the learning algorithm is summarized in Alg. 5.

The attack presented here is based on the Ahlberg–Nilson–Varah bound [38], which gives a control on the sup norm of a matrix with dominant diagonal elements. More precisely, when presented with a mini-batch $\mathcal{B}$, the attacker needs to modify the contexts and the rewards. We assume that the attacker knows the number of mini-batch updates $M$ and has access to a lower-bound on the reward of the target arm, $\nu$ as in Assumption 2.

The attacker changes $(K-1) \times d$ rows of the first mini-batch to rewards of $0$ with a context $\delta_a e_i$ for each arm $a \neq a^\dagger$ with $(e_i)$ the canonical basis of $\mathbb{R}^d$. Moreover, $\delta_a$ is chosen such that:

$$\delta_a > \max\left( \sqrt{\frac{2MBL^2 d}{\nu} + dMB}, \sqrt{\frac{4\beta_{max}^2 L^2 d}{\nu^2} + dMB} \right) \tag{13}$$

**Algorithm 5** Semi-Online Attack Setting.
---
**Input:** Bandit alg. $\mathfrak{A}$, size of a mini-batch: $B$
Set $t = 0$
**while** True **do**
    $\mathfrak{A}$ observe context $x_t$
    $\mathfrak{A}$ pulls arm $a_t$ and observes reward $r_t$
    Interaction $(x_t, a_t, r_t)$ is saved in mini-batch $\mathcal{B}$
    **if** $|\mathcal{B}| = B$ **then**
        Attacker modifies mini-batch $\mathcal{B}$ into $\tilde{\mathcal{B}}$
        Update alg. $\mathfrak{A}$ with poisoned mini-batch $\tilde{\mathcal{B}}$
    **end if**
**end while**
---

with $\beta_{max} = \max_{t=0}^{MB} \beta_a(t)$ and $M$ the number of mini-batch updates.

**Proposition 4.** *After the first attack, with probability $1 - \delta$, LINUCB always pulls arm $a^\dagger$,*

*Proof.* After having poisoned the first mini-batch $\mathcal{B}$, the latter can be partitioned into two subsets, $\mathcal{B}_c$ (with non-perturbed rows) and $\mathcal{B}_{nc}$ (with the poisoned rows). The design matrix of arm $a \neq a^\dagger$ for every time $t$ after the poisoning is:

$$V_{t,a} = \lambda I_d + \sum_{l=1,a_l=a}^{t} x_l x_l^\intercal + \delta_a^2 \sum_{i=1}^{d} e_i e_i^\intercal \tag{14}$$

For every time $t$, non diagonal elements of $V_{t,a} = (v_{i,j})_{i,j}$ are bounded by:

$$\forall i, r_i := \sum_{j \neq i} v_{i,j} \leq \sum_{j \neq i} \sum_{l=1,a_l=a}^{t} \|x_l x_l^\intercal\|_\infty \leq d N_a(kB) \tag{15}$$

Whereas for all diagonal elements, $v_{i,i} \geq \delta_a^2 > r_i$. Thus $V_{t,a}$ is strictly diagonal dominant and by the Ahlberg–Nilson–Varah bound [38]:

$$\|V_{t,a}^{-1}\|_\infty \leq \frac{1}{\min_i (\|v_{i,i}\| - r_i)} \leq \frac{1}{\delta_a^2 - dMB} \tag{16}$$

Then, for every arm $a \neq a^\dagger$ and any context $x \in \mathcal{D}$ and any time $t$ after the attack:

$$\langle \hat{\theta}_{a,t}, x \rangle + \beta_a(t) \|x\|_{V_{t,a}^{-1}} \leq \sum_{l=1,a_l=a}^{t} r_t (V_{t,a}^{-1} x_t)^\intercal x + \beta_a(t) \|x\|_1 \sqrt{\|V_{t,a}^{-1}\|_\infty}$$

$$\leq \|V_{t,a}^{-1}\|_\infty d N_t(a) \sup_{y \in \mathcal{D}} \|y\|_2^2 + \beta_{\max} \sqrt{d} \sup_{y \in \mathcal{D}} \|y\|_2 \sqrt{\|V_{t,a}^{-1}\|_\infty} < \nu$$

We have shown that for any arm $a \neq a^\dagger$ and any time step $t$ after the attack, the upper confidence bound computed by LINUCB is upper-bounded bu $\nu$ the arm $a^\dagger$. Then, with probability $1 - \delta$, the confidence set for arm $a^\dagger$ holds and, for all $x \in \mathcal{D}$, arm $a^\dagger$ is chosen by LINUCB. The total cost of this attack is $d \sum_{a \neq a^\dagger} \delta_a L = O(\sqrt{MB})$ $\qquad \square$