[Reviews · NeurIPS 2020]

Review 1

Summary and Contributions: Summary & Contributions: Authors study the scopes of adversarial attacks in linear contextual bandit algorithms which have applications in a wide range of domains. The authors consider adversarial attacks on both reward and the context and analyze the robustness (or lack of it) of various contextual linear bandit algorithms including LinUCB, LinTS, epsilon-greedy etc. Empirical evaluations are presented on various synthetic and two real datasets to examine the effect of attacks on these algorithms.

Strengths: The problem of analyzing the effect of adversarial attacks on bandit algorithms are indeed interesting and well motivated, and present work is supposedly the first one to analyze this for stochastic contextual linear bandits. Authors also analyze some popularly studied bandit algorithms, like LinUCB, LinTS, epsilon-greedy, and showed the attacking strategies (as optimization problems) to fool above algorithms for playing some targeted suboptimal arm majority number of times. Experiments are fairly detailed and reported on a large set of datasets showing the effect on learning rate of existing techniques on different degree+type of attacks.

Weaknesses: 1. Earlier works: While I like the problem formulation and inferred results, what concerns me is the earlier work by June at al, 2018 [19]. The reason being they just address the almost same problem for stochastic bandits, with exact same adversarial attack formulation on the rewards, and they provide a similar analysis on effect of LinUCB, LinTS, epsilon-greedy algorithms on such attacks (similar optimization objective based attacking strategy etc). So the present work only appears to be an extension of their problem formulation+analysis technique for contextual cases. Is there any unique contribution of this paper besides that. 2. Intuition behind two corruptions: Another contribution, claimed to be the first attempt of this paper, is studying the case for "context-corruption" (Sec 4). But I fail to see why is it fundamentally any different than corrupting the rewards (Sec 3), as rewards being a linear function of the context vectors, corrupting later is equivalent to corrupting the former and similar ideas are in fact used in the analysis of Sec 4. Please clarify if there is fundamental difference in the attacking strategy required for context corruption. 3. Robust algorithms: This more of a suggestion. Paper presently analyzes the strategy to corrupt the rewards (contexts) such that some well known techniques, like LinUCB, LinTS, epsilon-greedy, fail to learn the best policy, similar to the results + analysis of [19]. However what would have been really interesting and useful is to design a robust contextual bandit algorithm which could survive the attacks (up to some corruption threshold), basically gives a regret guarantee which would be proportional to degree of corruption of non-stationarity, similar to the guarantees of Besbes et al (2014), Luo et al (2018) etc. 4. Presentation: This is a minor issue, but the paper could have structured better, eg a precise contribution would have been easier to parse summarizing the inferences on LinUCB, LinTS, epsilon-Greedy for different types of corruptions (reward, context etc) -- a table would have been the best. Please avoid writing large paragraphs which are throughout in the paper (Pg 2,4,7), break them into small paragraphs and give a paragraph name so that the readers know apriori what to expect after reading it. I also find hard to find out the main claims, the key results of each section could be presented as theorems with some following remarks to clearly mention what the inferences are.

Correctness: Justified.

Clarity: The paper suffers from unclear claims and lack of to the point presentation of results. writing could be improved, see point#4 "Weakness".

Relation to Prior Work: Reasonably clearly mentioned.

Reproducibility: Yes

Additional Feedback: Please see comments in the "Weakness" section. Overall some of the results are interesting but it is hard to appreciate their originality given the work of Jun et al (2018), if the specific+new contributions of this work could be fairly justified over the former paper, I would be happy to reconsider the scores. =============== Post rebuttal ==================== I went through the rebuttal and I changed my score to weak accept: Some parts of the works align with the existing results so I still do not see any major contribution of this paper either conceptually or algorithmically, but the problem is interesting and the paper covers many attacking models with reasonable empirical evaluations, so I tend to accept the paper now given the raised concerns on the presentations are taken care of. Also please add a discussion on the different attack models (on contexts and rewards) and their implications on the regret performance.


Review 2

Summary and Contributions: The paper investigates the robustness of contextual bandits when attacker exists. The attacker aims at causing the contextual bandit to pull a pre-specified target arm almost every time while achieving low attack cost. The work complements the previous works on attacking bandits, and pushes the frontiers of the area.

Strengths: (1). The paper has strong theoretical results that prove that contextual bandits are vulnerable to adversarial attacks, and the attacker can strategically manipulate the rewards or contexts to force a set of target arms almost every iteration. Meanwhile, the attack cost is sublinear, which is not a surprising but a convincing result. (2). Many different attack settings are studied, which includes changing reward, contexts, and targeting a specific user. This is really thorough investigation into the problem. (3). The paper performs well-organized experiments, and the empirical results match the theory.

Weaknesses: (1). Is it truly necessary to assume that the reward lies in (0,1)? Seems like it is critical for the attacker to know a lower bound of the reward, so that he can always drag down the reward to 0 to make non-target arms seemingly inferior. What if no such lower bound information is known beforehand? In that case, will the attacker still be able to attack? Note that prior works do not assume such lower bound information. (2). When the attacker changes the contexts, what if the attacker targets multiple users instead of a single user? I think in that case the attacker may not be able to successfully change the selected arms simultaneously for all target users. I am wondering if the authors have thought about that?

Correctness: Yes

Clarity: Yes

Relation to Prior Work: Yes

Reproducibility: Yes

Additional Feedback:


Review 3

Summary and Contributions: The paper studied adversarial attack on linear bandits, where the adversary may poison either the reward feedback or observed context cause the contextual bandit into pulling a pre-specified target arm(s) linear times. Online data poisoning attack have been studied for stochastic bandits and this is the first paper that considers online attack of linear contextual bandits. The paper considers several settings including online attack on reward for a no-regret algorithm, online attack on context for LinUCB and offline attacks on a single context. The paper proposed attacking strategies for these scenarios and theoretically analyzed the feasibility and cost of the proposed methods.

Strengths: 1) This paper is the first to study online adversarial attack on linear bandits, which is an important problem following recent works of online attack on stochastic MAB and offline attack on linear bandits. 2) This paper thoroughly covered different attacking scenarios for linear bandit problem. 3) Theoretical proofs of proposed methods are sound. 4) Experimental result on simulation data and real-word datasets validated the effectiveness of the attacks.

Weaknesses: 1) Regarding the attack on rewards in Sec3: the theoretical analysis only holds for rewards with r_{min} = 0 (in Assumption 1) and cannot generalize to other choice of r_min. This means that the context $x$ is limited to intersections of half-spaces decided by all \theta_a. This strong assumption makes the problem easier as the environment is no longer the standard contextual bandits problem (which can choose any x within its norm bounded), and limits the contribution of the theoretical analysis. 2) A minor concern is the assumption that the attacker knows the value of r_{min} and r_{max}. I acknowledged the argument in the paper that r_{min} and r_{max} may be accessible by the attacker in some real-world applications, but this assumptions make the problem a much easier (even trivial) setting than the first adversarial attack on bandits paper by Jun et al. (NeurIPS 2018), which proposed an adaptive attack strategy without the need of knowing r_{min} and r_{max}. 3) Regarding the attack on contexts in Sec 4: in order to make sure the norm of attacked context is bounded by L (following Assumption 1 where the norm un-attacked contexts are bounded by L), the paper added another assumption on the norm of un-attacked contexts are bounded by $\nu L/2$ in Proposition 2. This means that there is still a difference between the norm of attacked contexts and un-attacked contexts which makes the step of clipping the norm seems not well motivated and not beneficial to the attacker at all.

Correctness: I checked most of the derivations and they are correct.

Clarity: The paper is well written and easy to follow.

Relation to Prior Work: The paper clearly discussed the related works in data poisoning attack on bandits and more generally, reinforcement learning.

Reproducibility: Yes

Additional Feedback: While the authors show that the proposed method empirically works for general r_{min} and r_{max} , it is important to affirmatively answer whether the proposed method can successfully attack general r_{min} and r_{max} cases, i.e., less or even no restrictions on the environment's choice over context $x$. It would also be helpful to justify the norm clipping step in Sec 4. -------------Post Rebuttal-------------------------- I agree with the authors' argument that it seems hard to avoid the assumption of bounded reward if not knowing what algorithm to attack. I suggest the authors to include the discussion of this intuition in the paper.

[Author Response · NeurIPS 2020]

We thank the reviewers for their insightful comments.

**[R4] Knowing $r_{\min}$ and $r_{\max}$:** Jun et al. (2018) [19] relaxes the assumption of knowing $[r_{\min}, r_{\max}]$ when studying
rewards attacks *only* for $\varepsilon$-greedy and UCB while we provide an efficient attack for any algorithms in the linear
contextual case. We could also build an attack on LinUCB only without knowing $[r_{\min}, r_{\max}]$ using $\tilde{r}^2$.

**[R2, R4] Bounded Rewards:** We need the assumption that rewards are bounded in $(0, 1)$ to prove a formal bound on
the total cost of the attacks for *any* no-regret bandit algorithm, otherwise we need more information about the attacked
algorithm. In practice, the second attack on the rewards, $\tilde{r}^2$, can be used in the case of unbounded rewards for *any*
algorithms. We could not prove a bound on the total cost of the attacks with $\tilde{r}^2$ because the reward process becomes
non-stationary under this attack. Thus, there is no guarantee that an algorithm like LinUCB will pull a target arm as the
proof relies on the environment observed by the bandit algorithm being stationary.

The assumption of bounded reward is not present in earlier works but the proofs provided by them do not address the
issue of non-stationarity in the attacked reward process. We discussed it with the authors of Liu & Shroff (2019) [27],
who also study attacks on rewards independent of the bandit algorithms. We were not able to find a solution to this
problem for either of our theorem or theirs.

To sum up, it is possible to construct an attack which does not assume bounded rewards but this comes at the price of a
formal proof of the total cost for the attacker or knowing the bandit algorithm. We observe empirically that the total
cost of attack is sublinear when using $\tilde{r}^2$.

**[R1] Fundamental differences between attacking the rewards and the contexts:** Perturbing the contexts is funda-
mentally different from perturbing the rewards for the following reasons:

• The attacker only modifies the context that is *shown* to the MAB algorithm. The true context, which is used to
compute the reward, remains unchanged. In other words, the attacker cannot modify the reward observed by
the MAB algorithm. Instead, the attack algorithm described in Sec. 4 fools the MAB algorithm by making the
rewards *appear* small relative to the contexts and requires more assumptions on the MAB algorithm.

• For the attack on rewards, the attacker modifies the reward after the MAB algorithm has chosen an arm. When
the context is attacked, the MAB algorithm will choose an arm based on the attacked context. Therefore the
attack may change the arm pulled at the time-step of the attack. It allows for offline attacks as studied in Sec. 5,
which are not doable in the case of attacks on rewards. Studying online attacks on contexts requires to have
some control over the arm that is pulled and makes this problem more complex. We needed to show that the
attack algorithm described in Sec. 4 does not change the arm pulled by the bandit algorithm.

**[R1] Difference with Jun et al. (2018) and attacks on contexts:** Our work is indeed inspired by the work of Jun et
al. (2018). The main difference is that we extend the setting of adversarial attacks in bandits to the contextual case
which is in general a much more complex setting than the classical MAB setting. In addition, we also:

• show that adversarial algorithms such as EXP4 can be fooled by our attacks on rewards with a sublinear total
attack cost (see App. D).

• consider the setting of attacks on contexts, which is fundamentally different as discussed above.

• introduce a more realistic attack setting where the attacker has very limited power and can only modify the
context associated to *a single user*, see Sec. 5.

**[R4] Norm of contexts:** Clipping the norm of the attacked contexts is not beneficial for the attacker. It means that the
attack for a specific context was violating the assumption used by the bandit algorithm that contexts are bounded by $L$.
Prop. 2 is here to provide a theoretical grounding for the proposed attack, but in practice the contexts are bounded by $L$.
We failed to convey this clearly in the paper and will correct this in the revised version. We show experimentally that,
when the unattacked contexts are bounded by $L$ and not just by $\nu L/2$, the attack algorithm enjoys a logarithmic total
cost of attack and fools the bandit algorithm into pulling an arm from the target set.

**[R2] Attacks on multiple users:** We did not consider attacking multiple users in the setting of Sec 5. It would have
interesting applications as an attacker could infect multiple users. In that case, the theoretical study of the feasibility
condition would be more complex as the attacks could significantly modify the behaviour of the MAB algorithm.

**[R1] Robust Algorithms:** We thank the reviewer for the references around this idea. Prop. 1 shows that under
Assumption 1 it is not possible to build an algorithm robust to a logarithmic perturbation of the rewards. Also, building
such algorithms for attacks on context, be it a unique one or all contexts, is not something we have investigated yet.

**[R1] Presentation:** We thank the reviewers for their advice on how to improve the presentation of the paper. The issues
mentioned will be rectified in the revised version of the paper.

[Meta-Review · NeurIPS 2020]

While there were some concerns raised regarding the contribution of the paper above Jun et al 2018, overall the reviewers appreciated the results and the setting in the paper. Some suggestions were made regarding improving the presentation of the paper. I recommend the authors to take those into account when preparing the camera ready.